# Neuron-specific isoform of PGC-1α regulates neuronal metabolism and brain aging

Dylan C. Souder[1,6], Eric R. McGregor[1,6], Josef P. Clark[1], Timothy W. Rhoads[2], Tiaira J. Porter[3], Kevin W. Eliceiri [4], Darcie L. Moore [3], Luigi Puglielli [1,5] & Rozalyn M. Anderson [1,5] ✉

The brain is a high-energy tissue, and although aging is associated with dysfunctional inflammatory and neuron-specific functional pathways, a direct connection to metabolism is not established. Here, we show that isoforms of mitochondrial regulator PGC-1α are driven from distinct brain cell-type specific promotors, repressed with aging, and integral in coordinating metabolism and growth signaling. Transcriptional and proteomic profiles of cortex from male adult, middle age, and advanced age mice reveal an aging metabolic signature linked to PGC-1α. In primary culture, a neuron-exclusive promoter produces the functionally dominant isoform of PGC-1α. Using growth repression as a challenge, we find that PGC-1α is regulated downstream of GSK3β independently across promoters. Broad cellular metabolic consequences of growth inhibition observed in vitro are mirrored in vivo, including activation of PGC-1α directed programs and suppression of aging pathways. These data place PGC-1α centrally in a growth and metabolism network directly relevant to brain aging.

Mitochondrial dysfunction has been proposed as a hallmark of aging[1] and changes in mitochondrial pathways are a key feature of calorie restriction (CR)[2], a dietary intervention that can extend lifespan in diverse species. The integrity of mitochondrial sensing and adaptation is fundamental to cellular resilience and functionality[3], and multiple lines of evidence now point to a role for mitochondria in age-related disease, including disorders as different as cancer, cardiovascular disease, and neurodegeneration[4,5]. The ubiquitously expressed transcriptional co-activator peroxisome proliferator-activated receptor gamma-coactivator 1 alpha (PGC-1α) has been described as the master regulator of mitochondrial function. PGC-1α is key in integrating and responding to diverse signals, ensuring the cellular metabolic response matches the prevailing conditions[6]. In cultured cells, modest PGC-1α overexpression leads not only to mitochondrial activation but extends to extra-mitochondrial processes such as redox and lipid homeostasis, chromatin maintenance,

cell cycle, growth, and structural remodeling[7]. Failure in maintaining brain energetics has been linked to neurodegenerative disorders[8], yet cell-type specific details of energy balance regulation and a precise role for PGC-1α have not been defined.

The activity of the PGC-1α protein is controlled by numerous upstream regulators, post-translational modifications, and cell-type and context-specific binding partners that fine-tune its co-activator function[6]. There are multiple isoforms of PGC-1α derived from distinct promoter regions of the *Ppargc-1a* gene[9]. The promoters are activated in response to different physiological stimuli[10,11], and the resulting protein isoforms appear to have both common and isoform-specific gene targets[9,10,12]. For example, in skeletal muscle, PGC-1α transcript variants expressed off the alternative promoter are selectively induced by resistance exercise, while the canonical promoter is activated by endurance exercise[10]. These differences in promoter activity also extend to tissue-specific differences in PGC-1α isoform expression, as

[1]Department of Medicine, SMPH, University of Wisconsin Madison, Madison, WI, USA. [2]Department of Nutritional Sciences, University of Wisconsin Madison, Madison, WI, USA. [3]Department of Neuroscience, University of Wisconsin Madison, Madison, WI, USA. [4]Department of Medical Physics, University of Wisconsin Madison, Madison, WI, USA. [5]GRECC William S, Middleton Memorial Veterans Hospital, Madison, WI, USA. [6]These authors contributed equally: Dylan C. Souder, Eric R. McGregor. ✉e-mail: rozalyn.anderson@wisc.edu

skeletal muscle, white adipose tissue, and liver display unique profiles of PGC-1α isoform abundance[9]. In the central nervous system, a third promoter >500 kb upstream of the canonical transcription start site is active and has been shown to respond to hypoxia[11,13]. It is unclear how functionally redundant the PGC-1α isoforms derived from the brain-specific promoter are with those derived from the canonical and alternate promoters, whether there is specificity in how they are regulated, and how the relative abundance among brain cell types might differ.

Although not validated in primary cells, we previously reported that glycogen synthase-kinase 3-beta (GSK3β) regulates mitochondrial energy metabolism via PGC-1α in glioma cells (H4) and phaeochromocytoma-differentiated neurons (PC12)[14]. GSK3β inhibition increased mitochondrial respiration and membrane potential and altered NAD(P)H metabolism, and these changes were associated with activation of Ppargc1a canonical and alternate promoters. Our previous studies in mice indicated that aging impacts mitochondrial and redox metabolism in the hippocampus in a region and cell-type-specific manner. These changes are linked to lower PGC-1α protein levels[15]. GSK3β and PGC-1α show the same pattern of expression in the hippocampus in mice and monkeys, and their relative abundance is reflected in the level of mitochondrial activity. Here, we determine the impact of aging on metabolism and PGC-1α. We investigate the role of PGC-1α in primary neurons and astrocytes and determine the extent to which GSK3β regulates PGC-1α expression driven from the three different promoters. We define the impact of GSK3β inhibition on neuronal metabolism and growth indices and validate the conservation of these effects in vivo. Together, these results suggest that PGC-1α plays a role in brain aging and that differential PGC-1α isoform expression and differences in GSK3β sensitivity allow for distinct regulation of metabolism among brain resident cell types.

## Results

### Aging impacts brain immune, inflammatory, and neuronal functional networks

To investigate aging across the lifespan, C3B6-F1 hybrid male mice were maintained on a fixed calorie intake to avoid obesity and maximize health during aging[15,16]. Three age groups were defined: adult (10 months), late-middle age (20 months), and advanced age (30 months). To gain a molecular perspective on brain aging, bulk RNA sequencing was conducted on cortical tissue (n = 5 per age group), resulting in 1.6 billion sequencing reads (~105 million per sample). After trimming, reads were aligned to the mouse genome, GRCm39, and approximately 48,000 transcripts from 18,000 genes were detected, identified, and quantified. Differential expression analysis among the three age groups identified age-sensitive genes across the adult lifespan (10 months vs. 30 months; 471 genes), including early (10 months vs. 20 months; 69 genes) and late (20 months vs. 30 months; 47 genes) life-stages (Figs. 1A and S1A; Supplementary Data 1). Gene Set Enrichment Analysis (GSEA)[17] analysis was conducted to identify functional pathways impacted by age (Supplementary Data 1). Aggregate data of age-responsive pathways are shown in the rank plot (Figs. 1B and S1B). Upregulated pathways included immune/inflammatory, proteome maintenance, and metabolic pathways. Downregulated pathways included primarily neuronal functional pathways.

Weighted gene correlation network analysis (WGCNA)[18] is an alternate approach that considers the entire transcriptome and identifies clusters of transcripts with a shared expression pattern. WGCNA identified 30 modules (Supplementary Data 2) that were then subject to pathway analysis via the Kyoto Encyclopedia of Genes and Genomes (KEGG). Of the 3 modules with high pathway-to-transcript ratios, the largest included immune and inflammatory pathways (magenta module, 467 transcripts, 42 pathways). The module was highly enriched for viral response, infectious response, B cell, T cell, and complement

pathways (Fig. 1C). Other pathways of interest included phagosome, lysosome, and endocytosis. The mice were maintained in Specific Pathogen Free conditions, indicating that the enrichment of immune and inflammatory pathways was not due to infection but more likely a reflection of immune deregulation and sterile inflammation. The age-related induction of viral infection pathways identified in the GSEA suggested that perhaps native DNA might be triggering the response, in particular since transposable elements (TEs) have been linked to age-associated diseases[19]. Of the 1146 TE transcripts detected in the RNA-Seq data set (Supplementary Data 3), 7 were differentially expressed at advanced age (Fig. 1D), with 3 of these showing significant differences in expression by late middle-age (Fig. S2). The second module of interest (steel blue module, 41 transcripts, 14 pathways) featured pathways that are integral to neuronal function (Fig. 1E), including synapse pathways and pathways involved in intercellular communication. The identification of immune, inflammatory, and neurological pathways responsive to aging and age-related conditions is consistent with prior studies[20,21] and indicates that the environment of the aging brain is distinct from that of mature adult mice and that processes involved in neuronal function may become compromised with age.

### Metabolism of brain aging is reflected in the transcriptome and proteome

Our prior studies using this same cohort of mice focused specifically on the hippocampus and identified age-related changes in mitochondrial activity and NAD(P)H redox metabolism that were life-stage, region, and cell-type specific[15]. Applying KEGG analysis to the third module identified via WGCNA (green-yellow module, 432 transcripts, 24 pathways) revealed an enrichment of metabolic pathways (Fig. 2A and Supplementary Data 2), including pyruvate metabolism, glycolysis, Krebs cycle, oxidative phosphorylation, and amino acid metabolism. Neurodegenerative disease pathways are also featured, including Alzheimer's disease, Parkinson's disease, and amyotrophic lateral sclerosis, along with autophagy and mitophagy pathways. The green-yellow module was significantly enriched for genes encoding mitochondrial proteins (Fig. 2B), with 11% of transcripts overlapping with the MitoCarta[22]. Two additional modules (cyan and salmon) were strongly positively correlated with the green-yellow module (correlation coefficients 0.501 and 0.654, respectively) (Fig. S3). Although no pathways were identified in the salmon module, the cyan module showed a neurodegenerative signature (Supplementary Data 2). Both cyan and salmon modules showed an enrichment of genes encoding mitochondrial proteins (16% and 10%, respectively) (Fig. 2C). These data align with prior work that suggests there is a metabolic component to neurodegenerative disease[23–25]. PGC-1α is the master regulator of nuclear-encoded mitochondrial genes encoded by the Ppargc1a gene. RT-PCR using primers that detect all transcript isoforms derived from Pparc1a identified significantly lower levels in mice of advanced age compared to young and middle-aged adult mice (Fig. 2D). These findings suggest that changes in mitochondrial functional pathways are a significant factor in brain aging, that metabolic and neurodegenerative pathways are coordinated in their response to aging, and that these changes are associated with lower PGC-1α expression.

Although transcriptional changes provide useful information, changes at the transcript level do not necessarily predict changes at the proteome level, and the overlap between profiling platforms is often low[26,27]. Proteins and nucleic acids were isolated from the same cortical specimen, with the organic phase used for protein isolation. Proteomic data was obtained by liquid chromatography-tandem mass spectrometry (LC-MS/MS). Across all samples, the detected proteome included 3578 quantified proteins, of which 2651 were annotated (Supplementary Data 4). Comparison across age groups identified 300 proteins as different in abundance between 30-month-old mice and 10-month-old mice (unadjusted p < 0.05) and 175 that

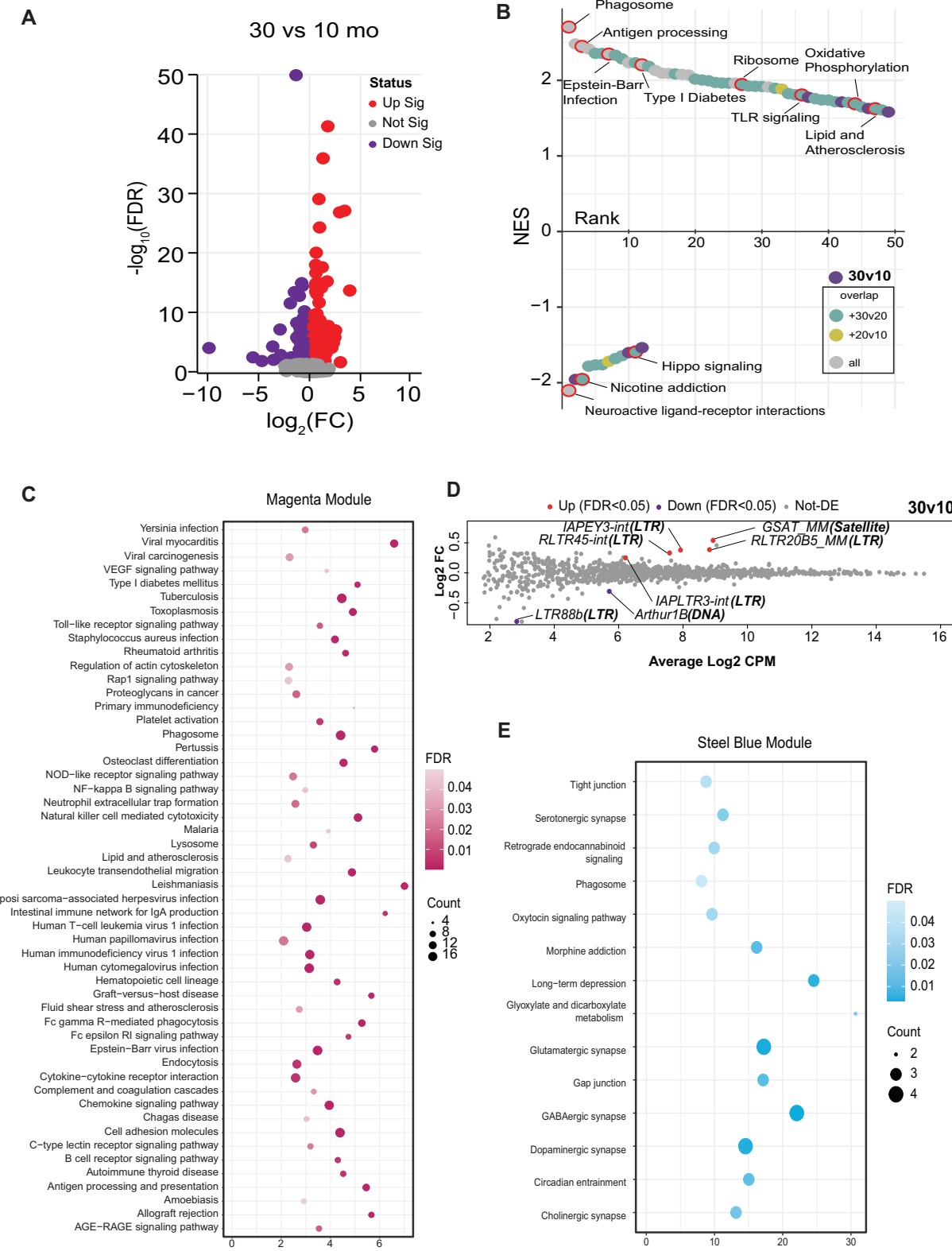

**Fig. 1 | Aging impacts brain immune, inflammatory, and neuronal functional networks. A** Volcano plot displaying transcripts quantified. Statistically significant transcripts are highlighted in red (upregulated) or purple (downregulated) in 30-month-old compared to 10-month-old male cortical tissue. **B** Rank order plot of enriched pathways by GSEA. Plot is ranked by normalized enrichment score. Each point represents pathways enriched in the 30v10 comparison. Points are color-coded by overlap with other comparisons (20v10–yellow, 30v20–cyan, 30v10 only–purple, present in all–grey). **C** KEGG pathways enriched in the Magenta module of WGCNA. **D** Mean-difference (MD) plots of transposable elements (TE) expression Log2FC against the average Log2 count-per-million (CPM) for 30 m/10 m. TE callouts: "TE name (**TE class**)." **E** KEGG pathways enriched in the Steel blue module of WGCNA. $n = 5$ mice/ age group.

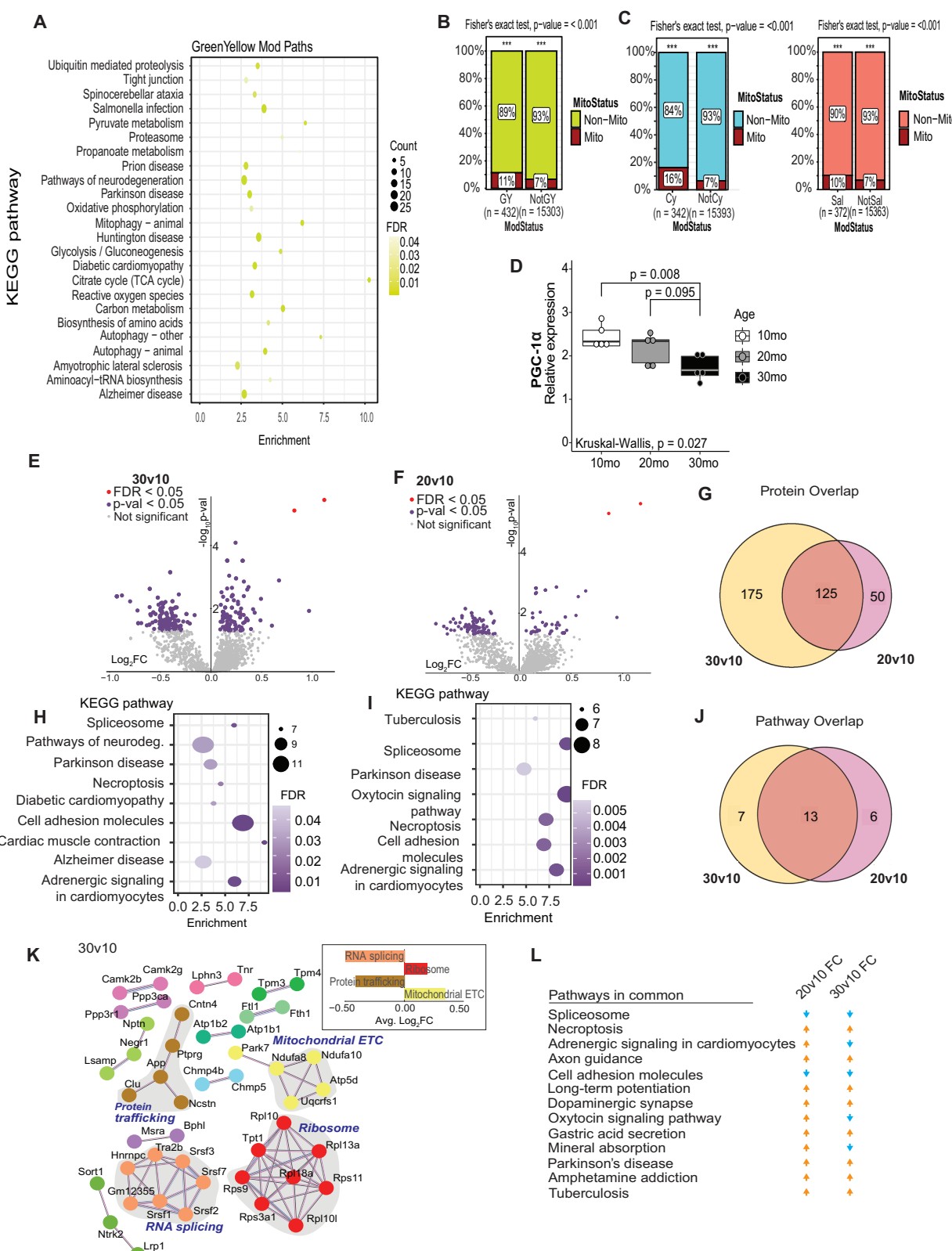

differed between 20-month-old and 10-month-old mice (Fig. 2E, F). Of those proteins that were responsive early during aging, ~70% persisted through the later aging phase (Fig. 2G). Functional enrichment of the protein differences for each comparison was assessed via overrepresentation analysis (ORA) using KEGG (Fig. 2H, I). Pathway overlap between age groups was 50% (Fig. 2J), including those associated with neurodegeneration (Parkinson's and

Alzheimer's, both of which include oxidative phosphorylation components), cell adhesion, adrenergic signaling, and RNA processing (spliceosome) (Fig. 2I, J). Taking an alternate approach, connections among age-sensitive pathways were also investigated via string (Fig. 2K). Functional enrichment analysis identified mitochondrial electron transport chain (ETC) and ribosomal pathways that were positively enriched and protein trafficking and RNA splicing

**Fig. 2 | Metabolism of brain aging is reflected in the transcriptome and proteome. A** KEGG pathways of the green-yellow module of WGCNA from mouse cortex. **B, C** Abundance of Mitocarta3.0 genes in the green-yellow ($p = 0.0004$) (**B**), cyan ($p < 0.0001$), and salmon ($p < 0.0001$) (**C**) modules. Signficance determined by two-sided Fisher's exact test. **D** RT-qPCR detection of PGC-1α expression cortex of male mice at 10 months, 20 months, and 30 months old ($n = 5$ mice/ age group). Boxplot represents median, 25th and 75th percentiles, and whiskers extend to min and max. Significance determined by Wilcox test (individual group comparison) and Kruskal–Wallis test. **E, F** Volcano plots displaying proteins quantified. Statistically significant proteins determined by Tukey's HSD test are highlighted in purple. **G** Venn diagram showing overlap between differentially expressed proteins in the 30-month-old or 20-month-old cortex compared to the 10-month-old. **H, I** KEGG pathways enriched in the 30-month-old cortical proteome (**H**) and 20-month-old cortical proteome (**I**). Only pathways with greater than 6 proteins were plotted. **J** Venn diagram showing the overlap between enriched KEGG pathways in the 30-month-old or 20-month-old cortex compared to the 10-month-old cortex. **K** Network identified by String analysis. Proteins are colored based on biological pathway—mitochondrial ETC (yellow), ribosome (red), protein trafficking (brown), RNA splicing (orange). **L** Table displaying common pathways between the 20v10 and 30v10 comparison and if they are activated or repressed compared to the 10-month-old. $n = 5$ mice/ age group.

pathways that were significantly downregulated in response to age (Tukey's, nominal $p < 0.05$). Finally, changes from the early to later aging phases were largely congruent in the directionality of change (Fig. 2L), indicating that at least some of the age-associated changes in the brain at the protein level are progressive and are initiated between 10 and 20 months of age. Overall, proteomics analysis demonstrates that aging-induced changes in mitochondrial pathways and are coincident with changes in processes linked to either growth or maintenance of neuronal function.

## PGC-1α transcript isoforms in the brain and cell type specificity

The brain is unique in producing variants of PGC-1α driven from a distal promoter located ~590 kb upstream from the canonical promoter of the *Ppargc-1a* gene (Fig. 3A). The brain promoter contains up to five brain-specific exons spliced to canonical exon 2 of the PGC-1α gene[13]. All possible isoforms from the brain promoter can be detected simultaneously using primers against brain exon 1 and canonical exon 2 (Supplementary Data 5), with the product hereafter referred to as PGC-1α B1E2. The isoforms generated from this transcript are predicted to encode proteins almost identical to that generated from the canonical promoter, PGC-1α (PGC-1α1), with small differences appearing only at the N terminus (Fig. S4A). In cortices from 1-year-old C3B6-F1 hybrid male mice, isoform-specific primers detected PGC-1α B1E2 transcript at ~eightfold greater abundance than canonical full-length PGC-1α1 and ~4.5-fold at greater abundance than that driven from the alternative promoter, PGC-1α4 ($n = 6$, $p < 0.0001$) (Fig. 3B). To understand how isoform expression in bulk tissue corresponded to cell types, primary neurons and primary astrocytes were isolated from postnatal mouse brains, cultured, RNA extracted, and PGC-1α isoforms were detected. Remarkably, neurons and astrocytes differed completely in PGC-1α expression profiles. Neurons displayed ~eightfold greater levels of PGC-1α B1E2 than PGC-1α1, with an intermediate expression of PGC-1α4 (Fig. 3C). PGC-1α B1E2 was not expressed in astrocytes; instead PGC-1α1 was the most abundant isoform, with a minor contribution from PGC-1α4 (Fig. 3C). To in vivo validate this finding, cell type-specific read density along the Ppargc1a locus was quantified using publically available RNA-seq data (GSE52564). Reads from the brain-specific exons (B1 and B4) were only detected in neurons and not in astrocytes, confirming that the B1E2 isoform is neuron-specific (Fig. 3D). The differential expression from the brain-specific, alternate, and canonical promoters between cell types can be seen in the normalized read counts (Fig. S4). Isoform-specific primers were used to probe mRNA extracted from cortex of mice aged 10 months, 20 months, or 30 months, and showed that PGC-1α1, PGC-1α4, and PGC-1α B1E2 all decrease with age ($p < 0.05$ for all isoforms) (Fig. 3E).

Neurons and astrocytes share a common progenitor; however, they differ in the expression of B1E2, suggesting that the brain promoter must be activated explicitly at some point during neuronal differentiation. To define when this happens, hippocampal neural stem cells (NSCs) were isolated, differentiated, and sampled at intervals over the 14-day differentiation process. To differentiate the NSCs, growth factors were removed, and the relative abundance of neurons in the differentiated culture was determined by qPCR. Beta-tubulin III, a marker of differentiated neurons, increased ~tenfold by day 7 of differentiation (Fig. S4). PGC-1α B1E2 was barely detected in NSCs, increased ~12-fold by day 3 of differentiation, and persisted at an elevated level for the remainder of the differentiation process (Fig. 3F). In contrast, PGC-1α1 levels were reduced nearly twofold by day three and recovered by day 14, with a similar detected for PGC-1α4 over this same time frame (Fig. S4). Primary neurons isolated from neonate and prenatal brains require several days in culture to fully mature and to express classic neuronal transcriptional signatures. To understand the dynamics of PGC-1α isoform expression during primary neuron maturation, neurons were isolated from P0 mouse cortices and monitored over 14 days. On the initial day of isolation (DIV 0), PGC-1α4 was nearly undetectable, while PGC-1α B1E2 was fourfold lower than PGC-1α1 (Fig. 3G). During maturation, PGC-1α1 did not change, increased expression of PGC-1α4 was detected by day 5 of maturation, and increased expression of B1E2 at day 10 (Figs. 3H and S4). To determine if there were differences in stability of the neuronal transcript isoforms of PGC-1α, primary neurons were treated with transcription inhibitor actinomycin D. After 24 h, PGC-1α1 levels were reduced by 85%, PGC-1α B1E2 was decreased by 58%, and PGC-1α4 was reduced by 43% (Fig. 3I). The basis for stability differences is unclear, particularly between PGC-1α1 and B1E2, which differ only in the 5' region. These data suggest that neuronal PGC-1α transcript isoforms are not regulated equivalently or in parallel.

To understand if differences in PGC-1α isoform expression between primary neurons and primary astrocytes correlated with differences in cellular metabolic status, fixed cells were analyzed by fluorescence lifetime imaging microscopy (FLIM). This method allows for an independent assessment of the relative contribution of bound and free NAD(P)H to total NAD(P)H pools by fitting the decay curve to fast ($t_1$) and slow ($t_2$) photon release that are associated with free and bound fluorophores respectively (Fig. 3J, K). Mean fluorescence lifetime ($\tau_m$) was significantly higher in neurons than in astrocytes (Fig. 3L). Analysis of lifetime components of NAD(P)H revealed that the neuronal decay curve ($\tau_m$) was comprised of a greater contribution from free than bound cofactors (70/30 ratio), but the decay curve in astrocytes had an even greater contribution (80/20) from free cofactor compared to bound (Fig. S5). These data suggest that there may be greater reliance on the TCA cycle and electron transport chain (protein-bound NADH) in primary neurons compared to astrocytes, an outcome that was not unexpected[28]. Total fluorescence intensity, a measure of cellular NAD(P)H, was higher in neurons than in astrocytes for both nuclear and cytosolic pools (Fig. 3M). Taken together, these data show brain that PGC-1α transcript expression changes with age across all three promoters, that PGC-1α B1E2 emerges through differentiation and maturation as neurons commit to their fate, and that cell-type differences in PGC-1α isoform expression between neurons and astrocytes are linked to distinct metabolic states.

## PGC-1α gene promoter activation by GSK3β and associated factors

In cultured cells, PGC-1α activity and protein stability can be modulated through the growth-sensitive kinase GSK3β[14,29]. In mice,

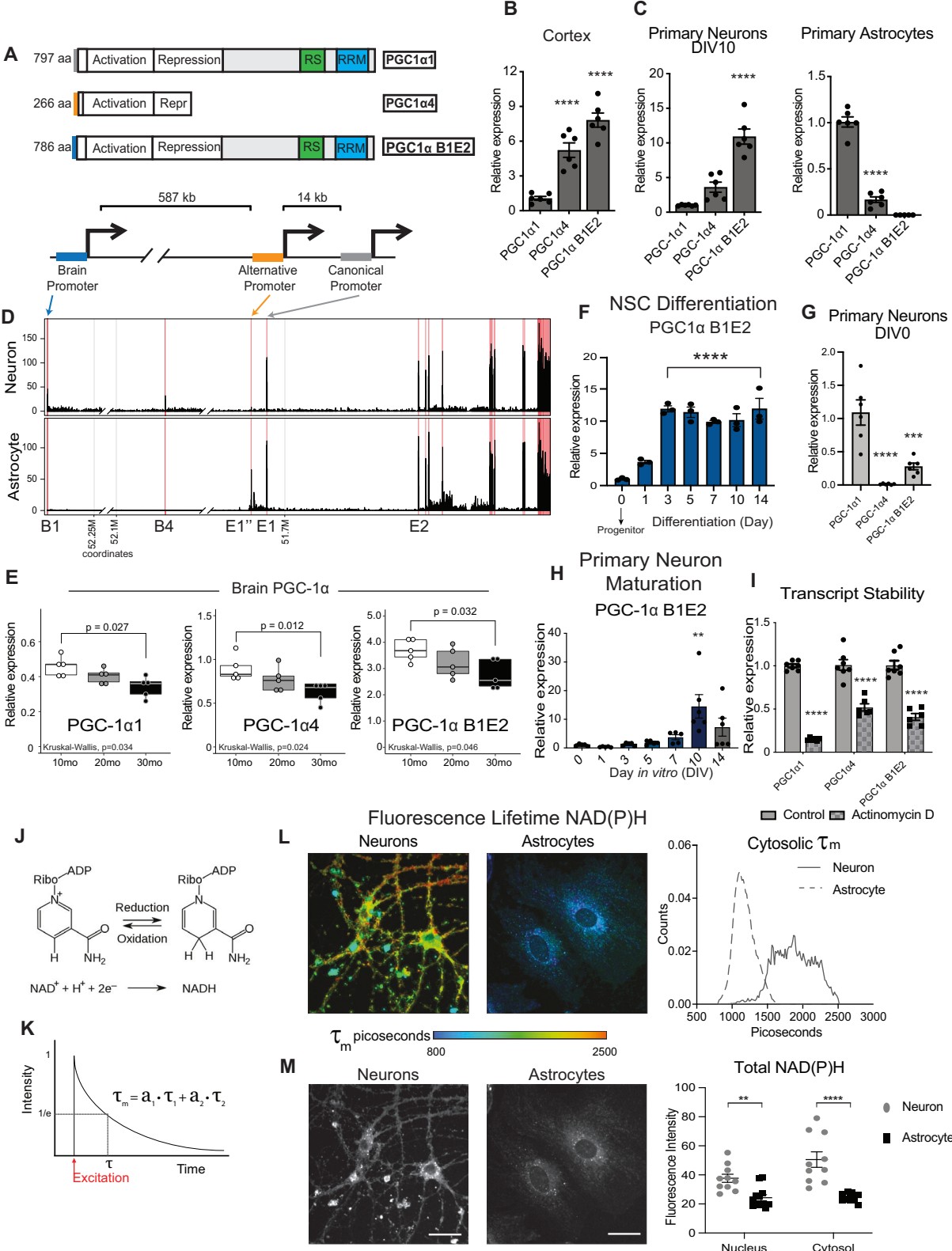

immunodetection of GSK3β expression in the hippocampus from the same 10, 20, and 30 months of age mice revealed significant age and region effects where GSK3β was increased at the later age phase (Fig. S6, 2-way ANOVA $p < 0.01$). In glioma cells (H4) and phaeochromocytoma-differentiated neurons (PC12), GSK3β inhibition using lithium chloride (LiCl) causes an increase in expression of PGC-1α1 and PGC-1α4 isoforms[14], but neither cell type expresses the B1E2

isoform which is the dominant form of PGC-1α in primary cortical neurons. Primary neurons were treated with lithium chloride for 24 h, and isoforms of PGC-1α were detected by RT-PCR. Unexpectedly, B1E2 expression from the brain promoter was suppressed by LiCl treatment even though, consistent with our prior reports, PGC-1α1 and PGC-1α4 transcripts were induced (Fig. 4A). Changes in isoform expression were muted in primary cortical astrocytes, with a trend toward

**Fig. 3 | PGC-1α transcript isoforms in the brain and cell type specificity.**
**A** Schematic of the three major isoforms of Ppargc1a expressed in the brain and a representation of the location of their distinct promoter regions. **B, C** Relative expression pattern of PGC-1α transcripts detected by RT-qPCR of 12-month-old male mouse cortex ($n = 6$ mice) (**B**) and DIV10 primary cortical neurons and astrocytes isolated from P0 (neurons) and P1 (astrocytes) neonates ($n = 6$) (**C**). **D** Read density from GSE52564 for isolated neurons and astrocytes across the Ppargc1a locus. Pink lines denote annotated exons. **E** RT-qPCR detection of Ppargc1a transcript variants in cortex of male mice at 10 months, 20 months, and 30 months old ($n = 5$ mice). Boxplot represents median, 25th and 75th percentiles, and whiskers extend to min and max. **F** Expression of the brain-specific isoform of PGC-1α during neural stem cell (NSC) differentiation ($n = 3$ biological replicates). **G** Relative expression pattern of PGC-1α transcripts detected in DIV0 P0 neurons by RT-qPCR ($n = 6$ biological replicates). **H** Detection of Ppargc1a B1E2 during maturation of P0 primary cortical neurons (DIV0, 1, 3, 5, 7: $n = 5$; DIV10, 14: $n = 6$).

**I** PGC-1α transcript levels after 24 h of actinomycin D treatment ($n = 7$). **J** Schematic of oxidation-reduction reaction of $NAD^+$ to NADH. **K** Example of two-component decay curve produced through fluorescence lifetime imaging microscopy and is represented by the equation $\tau_m = a_1(\tau_1) + a_2(\tau_2)$. **L** Representative images (left) and distributions (right) of NAD(P)H mean fluorescence lifetime images of primary neurons and primary astrocytes ($n = 10$ neurons, $n = 12$ astrocytes). **M** Representative images (left) and quantitation (right) of NAD(P)H fluorescence intensity of primary neurons and primary astrocytes ($n = 10$ neurons, $n = 12$ astrocytes). Data shown with error bars noting mean ± SEM (**B, C, F–I, M**). Signficance determined by ordinary one-way ANOVA with Dunnett's multiple comparisons test (**B, C, F–H**), the Wilcox test (individual group comparison) and the Kruskal–Wallis test (**E**), or by two-way ANOVA with Sidak's multiple comparisons test (**I, M**). Asterisks indicate $p$ value of <0.05 (*), <0.01(**), <0.0001 (****). See Source data file for exact $p$-values. See methods section for details on biological replicates used for RT-qPCR.

---

increased expression in PGC-1α4 detected (Fig. S7). These data suggest that mechanisms controlling the expression from the various Pparg1a promoters are not equivalent and differ by cell type.

To investigate potential differences in mRNA stability, neurons were treated for 24 h with LiCl in the presence or absence of Actinomycin D. If mRNA stability were to explain the observed effect of LiCl then the expectation would be PGC-1α1 and PGC-1α4 stabilized and B1E2 de-stabilized, but this is not what was observed. LiCl modestly increased the stability of PGC-1α1 and B1E2 with no significant change in the truncated alternative isoform PGC-1α4 (Fig. 4B). Next, a candidate approach was used to identify factors that might contribute to LiCl-directed regulation of Ppargc1a gene expression. Inhibitory phosphorylation of GSK3β (S9) was increased within 24 h of treatment, as expected. LiCl also induced activating phosphorylation of CREB (S133), a transcription factor known to regulate PGC-1α expression[30,31], and AMPK (T172), another PGC-1α activator (Fig. 4C). Prior studies had placed CREB downstream of GSK3β[32], but a connection between AMPK and GSK3β has not been described. Analysis of canonical CREB binding motifs at each of the PGC-1α promoters revealed multiple putative CREB binding sites in each of the promoter regions of the *Ppargc-1a* gene (Fig. 4D), arguing that LiCl may act in part through CREB. These and other factors were tested for their role in LiCl-directed regulation of PGC-1α (Fig. 4E). LiCl is not a specific GSK3β inhibitor[33], so to directly test for GSK3β regulation of PGC-1α, neurons were treated with a specific inhibitor called inhibitor VIII. Treatment with inhibitor VIII phenocopied the effects of LiCl on PGC-1α4 (activated) and on B1E2 (repressed) but not on PGC-1α1 that only responded to LiCl and not inhibitor VIII (Fig. 4F). Inhibition of CREB using inhibitor 666-15 had no impact on basal PGC-1α expression across all three isoforms but blunted the activation of PGC-1α1 and PGC-1α4 transcripts in response to LiCl, indicating that CREB is required. LiCl and CREB inhibition appeared to have an additive effect in repressing B1E2 (Fig. 4G), suggesting that additional factors signaling to CREB are also involved. Treatment of neurons with an AMPK inhibitor, compound C, blocked the activation of PGC-1α1 and PGC-1α4, but the GSK3β-directed repression of B1E2 was not dependent on AMPK (Fig. 4H). Going upstream in the signaling pathway, neurons were next treated with ANA-12, a TrkB inhibitor. Increases in expression in PGC-1α1 and PGC-1α4 were blunted but not entirely blocked, while there was no impact on B1E2 (Fig. 4I). Many transcriptional regulators are regulated not at the level of transcription but by post-translational modification and subcellular localization. Nonetheless, targeted analysis of lithium-responsive genes identified several transcriptional repressors, co-repressors, and chromatin-modulating factors such as Per2 and Phf12 associated with circadian regulation and EZH2 and EED associated with Polycomb complex mediated repression (Fig. 4J). Taken together, these experiments show that the canonical and alternative promoters are regulated similarly in response to LiCl, requiring TrkB, CREB, and AMPK for full implementation of the response to GSK3β inhibitors;

however, the brain-specific promoter seems to be controlled by an alternate mechanism downstream of GSK3β inhibition involving an unidentified factor.

### Neuronal metabolic response to GSK3β inhibition with Lithium

The consequences of the PGC-1α B1E2 isoform repression were investigated further via RNA-seq of LiCl-treated primary cortical neurons compared to untreated controls ($n = 4$ per group). Transcripts representing 15,370 genes were identified. Five thousand four hundred eighty were identified as differentially expressed using an FDR of 0.01. Top LiCl-responsive genes included ATF3, Aquaporin 9, Slc20a3 (synaptic vesicle-linked zinc transporter), and homeobox transcription factors (Supplementary Data 6). Differentially expressed genes were subject to pathway analysis via GSEA (Fig. 5A). Enriched and activated pathways included those associated with growth, such as cell cycle, Hippo signaling, and PI3K-Akt signaling that is part of the insulin and mTOR signaling pathway. Ordinarily, GSK3β is inhibited by growth signaling, but these data indicate that GSK3β inhibition as an initiating event might stimulate growth pathways, at least in neurons. The most highly enriched and suppressed pathways include ribosome, oxidative phosphorylation, proteosome, and several neurodegenerative disease pathways. Notably, many pathways that LiCl repressed were among those activated by aging in the brain, including oxidative phosphorylation (Fig. 5B) and ribosome (Fig. 5C). Focusing on established targets of PGC-1α transcriptional co-activation, a suite of genes associated with the electron transport chain (Cox4i1, Ndiufb8, Cycs, Vdac1, Cox5b), TCA cycle (Sdhb, Idh3a), and antioxidant maintenance (Sod2 and Cat) appear downregulated in response to LiCl, while expression of other PGC-1α associated factors including Nrf1, Ppara, Ucp2 were induced (Fig. 5D). Genes associated with the neuronal function (Syt1 and Cplx1) appear to be the most down-regulated, supporting PGC-1α B1E2's role in modulating neuron-specific functions[12].

Changes in gene expression suggested that LiCl would blunt oxidative metabolism in neurons. Oxygen consumption was quantified continuously in primary neurons treated with LiCl using the RESIPHER oxygen consumption monitor. By 24 h, the oxygen consumption of the neurons in culture was significantly lower than that of untreated neurons (Fig. 5E). Neurotrophic factors (NGF, VEGFa, BDNF) were modestly but significantly induced in response to lithium; however, the physiological significance of these changes is unclear in the context of isolated primary neurons (Fig. S8). The glycolytic pathway was not identified as being enriched in our RNASeq analysis. Extracting the transcripts associated with the Glycolysis GO term showed that the rate-limiting enzyme phosphofructokinase (gene: PFKM) was lower in expression in lithium treated neurons and levels of the fructose 2, 6, bisphosphatase/6 phosphofructo-2-kinase (gene: Pfkfb2) that produces fructose 2,6, bisphosphate (positive allosteric regulator) was also lower (Fig. S8). These gene expression changes would argue for a

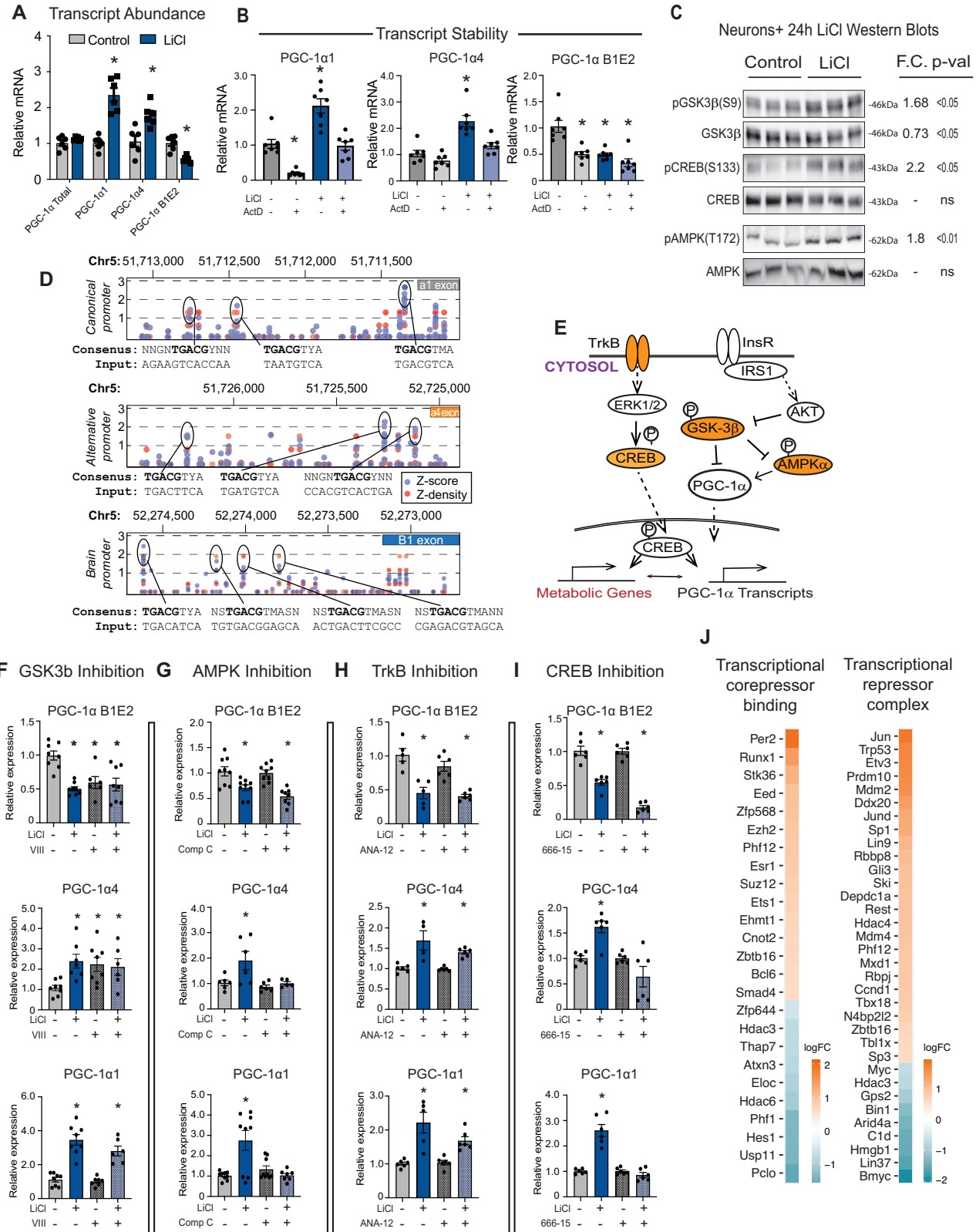

dampening of metabolic activity rather than a switch between glycolysis/FAO or respiring/nonoxidative metabolism.

Mitochondrial membrane potential was also lowered by 24 h LiCl treatment. An equivalent reduction was detected in primary neurons treated for 24 h with Inhibitor VIII, a GSK3β-specific inhibitor (Fig. 5F). Morphology was assessed by immunofluorescent detection of TOMM20 (translocase of outer mitochondrial membrane 20). Both

lithium and Inhibitor VIII exposure (24 h) resulted in significantly lower mitochondrial size (Fig. 5G) and integrated density (Fig. S9A), while the circularity index of the mitochondria was increased (Fig. S9B). Fluorescence lifetime imaging of redox cofactors NAD(P)H using 2-photon microscopy detected a slight decrease in mean fluorescence lifetime ($\tau_m$) in primary neurons treated with LiCl with an increased contribution from free NAD(P)H pools, consistent with the cells having lower

**Fig. 4 | PGC-1α gene promoter activation by GSK3β and associated factors.**
**A** RT-qPCR detection of PGC-1α transcripts in control and LiCl-treated neurons
($n = 6$ replicates). **B** RT-qPCR detection of PGC-1α transcripts after 24-h treatment
with actinomycin D +/− LiCl ($n = 7$ replicates). **C** Immunoblots of phosphorylation
and total protein of GSK3β, CREB, and AMPK after 24-h treatment with 15 mM
lithium chloride (LiCl) ($n = 6–7$). **D** Prediction of CREB binding sequences in Mmu
Ppargc1a promoters using a transcription factor binding motif prediction software
(http://tfbind.hgc.jp/). **E** Schematic of proposed pathways of lithium regulation of
Ppargc1a transcripts (orange−proteins investigated in (**F−I**). **F−I** RT-qPCR detection
of Pparc-1a transcripts following treatment with lithium in the presence or absence
of GSK3β inhibitor VIII ($n = 6–8$) (**F**), AMPK inhibitor Compound C ($n = 5–9$) (**G**),
TrkB inhibitor ANA-12 ($n = 4–6$) (**H**), or CREB inhibitor 666-15 ($n = 6$) (**I**) (see Source
data file for specific $p$-values for (**F−I**). **J** Heatmap of the significantly changing genes
detected from the "Transcriptional corepressor binding" and "Transcriptional
repressor complex" GO terms. Data shown with error bars noting mean ± SEM
(**A**, **B**, **F−I**). Asterisk (*) indicates $p$-value $< 0.05$ by two-way ANOVA with Sidak's
multiple comparison test (**A**), ordinary one-way ANOVA with Dunnett's test (**B**, **F−I**),
or two-tailed unpaired student's $t$-test (**C**). See Source data file for exact $p$-values
and for exact n for each group of (**F−I**). See methods for details on biological
replicates used for RT-qPCR and western blots.

reliance on oxidative metabolism (Figs. 5H and S9C). Additionally, a
decrease in cytosolic NAD(P)H fluorescence intensity was detected in
primary neurons treated with LiCl, consistent with the cells having
lower oxidative metabolic activity (Fig. 5I). Biochemical assessment of
redox ratios for NAD and NADP in LiCl-treated whole cell extracts are
consistent with the FLIM results. In enzyme-linked assays of NAD
cofactor, significantly lower levels of the reduced form, NADH, were
detected, with no change in the oxidized form, NAD+, resulting in a
higher redox ratio (Figs. 5J and S9D). For NADP cofactor, NADP+ and
NADPH levels were both lower in LiCl-treated neurons, but the redox
ratio was unchanged. These data point to a change in the redox state of
cells treated with LiCl and connect redox metabolism to GSK3β. The
metabolic impact of LiCl in primary neurons contrasts with that of
other cell types that do not express PGC-1α B1E2[14]. In all cases, LiCl
induces transcripts from the alternate and canonical promoters, but
the repression of expression from the brain promoter is unique to
neurons. The effect of LiCl on lower levels of mitochondrial-associated
metabolic processes in neurons points to a functional dominance of
PGC-1α B1E2 above other isoforms and suggests a means to confer cell-
type specificity in the metabolic response to growth via regulation of
PGC-1α.

## GSK3β plays an integrating role in growth regulation
Prior studies in suggested that growth and metabolism networks
might be interconnected via the GSK3β/ PGC-1α axis[14], but the effect in
primary neurons was not investigated. Here, treatment with LiCl
resulted in higher levels of inhibitory phosphorylation of insulin
receptor substrate -1 (IRS-1) in the phosphatidyl inositol 3 kinase (PI3K)
binding domain (S632) and lower levels of activating phosphorylation
of AKT (T308) and S6 ribosomal protein (S240) (Fig. 6A). These dif-
ferences in post-translational modification are consistent with the
repression of the insulin/IGF and mTOR Complex 1 signaling pathways.
Although levels of activating phosphorylation of ERK1/2 were
numerically higher in LiCl treated neurons the difference was not
significant, arguing that some but not all aspects of growth regulation
are responsive to lithium. These data and the activation of CREB and
AMPK (Fig. 4C) place GSK3β at the center of an interwoven growth and
metabolism regulatory network (Fig. 6B).

The transcriptional effect of LiCl was investigated using RNA-seq
of primary neurons treated for 24 h. Cell Cycle was the top-ranked
enriched pathway (Fig. 6C). At first glance, this might be an unex-
pected outcome since primary neurons are terminally differentiated;
however, many of the genes within that pathway are also involved in
cytoskeleton rearrangement, DNA surveillance, repair and main-
tenance, and chromatin remodeling (Supplementary Data 5). The
second-ranked pathway was Nucleo-cytosolic Transport, including
structural and transporting components, indicating that LiCl induces a
nuclear response not limited to changes in gene expression (Fig. 6D).
Under microscopy, LiCl caused changes in neuronal morphology, an
observation supported by the identification of the ECM-receptor
pathway among those highly enriched (Fig. 6E). Within this pathway, 9
of the top 10 genes have been linked to neuronal outgrowth, while the
other, syndecan 1 (Sdc1) is involved in neuronal progenitor

maintenance and proliferation[34]. Immunodetection of tubulin net-
works in untreated and LiCl-treated primary neurons revealed sig-
nificant changes in cell shape and length and in dendritic branching
that were quantified using Sholl analysis (Fig. 6F). Changes in cell
morphology were not associated with toxicity or cell death (Fig. S9E).
These data show that GSK3β inhibition is associated with changes in
intracellular growth signaling and pathways related to growth, that this
signature is transmitted to nuclear processes, and that growth signal-
ing has functional consequences for neuronal morphology.

To understand whether the impact of lithium on metabolism and
growth might be conserved in vivo, B6C3-F1 hybrid male mice were fed
a diet supplemented with lithium for 4 months[14]. Lithium carbonate
($Li_2CO_3$; 0.6, 1.2,1.8, or 2.4 g/kg) was incorporated into the diet of
individually housed mice fed daily equivalent amounts of food starting
at 2 months of age (Fig. 6G). Food intake was set by design to
equivalence for all animals (single housed, ~95% ad libitum food intake
provided daily, records of food intake maintained) and confirmed over
the duration of the experiment. At the higher doses (1.8 and 2.4 g/kg),
mice gained less weight than their counterparts that were untreated or
on lower doses (Fig. 6H). Body composition analysis via DEXA (dual
x-ray absorptiometry) showed that differences in body weight were
explained entirely by differences in adiposity (Fig. 6I). $Li_2CO_3$ treat-
ment dampened growth at the whole organism level independent of
food intake, favoring a reduction in adiposity without compromising
lean mass. These data indicate that the connection between metabo-
lism and growth via GSK3β identified in cultured cells may also be
relevant at the systems level.

## Lithium impacts brain growth and metabolism pathways
Differences in hippocampal mitochondrial and redox metabolism in
response to lithium carbonate treatment have been reported
previously[14]. The metabolic response to lithium was associated with
dose-dependent increases in GSK3β phosphorylation and increased
PGC-1α protein, although the effects were highly region and dose
specific. The cortical transcriptome was investigated via bulk RNA-Seq,
($n = 4$ per group) (Fig. 6G). The impact of $Li_2CO_3$ was dose-dependent,
with increasing numbers of responsive transcripts detected as the
dose increased, with 15, 91, 176, and 358 responsive genes for each of
the 4 doses, respectively (Fig. 7A and Supplementary Data 7). Pathway
analysis via GSEA using an FDR cutoff of 0.1 identified few pathways for
any dose. PathfindR is an alternate approach that identifies active
subnetworks of altered genes before performing the enrichment
analysis[35]. A greater number of lithium-responsive pathways was
identified with each step up in drug dose, with 61, 108, and 126 for
doses 1.2, 1.8, and 2.4 g/kg, respectively (Fig. 7B and Supplementary
Data 7). There was only one pathway in the low-dose (0.6 g/kg) axon
guidance, which was detected in the analysis of all other doses.

Unlike the case with aging, where genes were not uniformly
changing across age groups, for this cohort, by and large, there was a
progressive, consistent change in levels of responsive genes across
drug doses in these young animals (Fig. 7C). The overlapping genes
among all doses included central transcription regulators responsive
to growth, Fos and Junb and early growth response (EGR) genes that

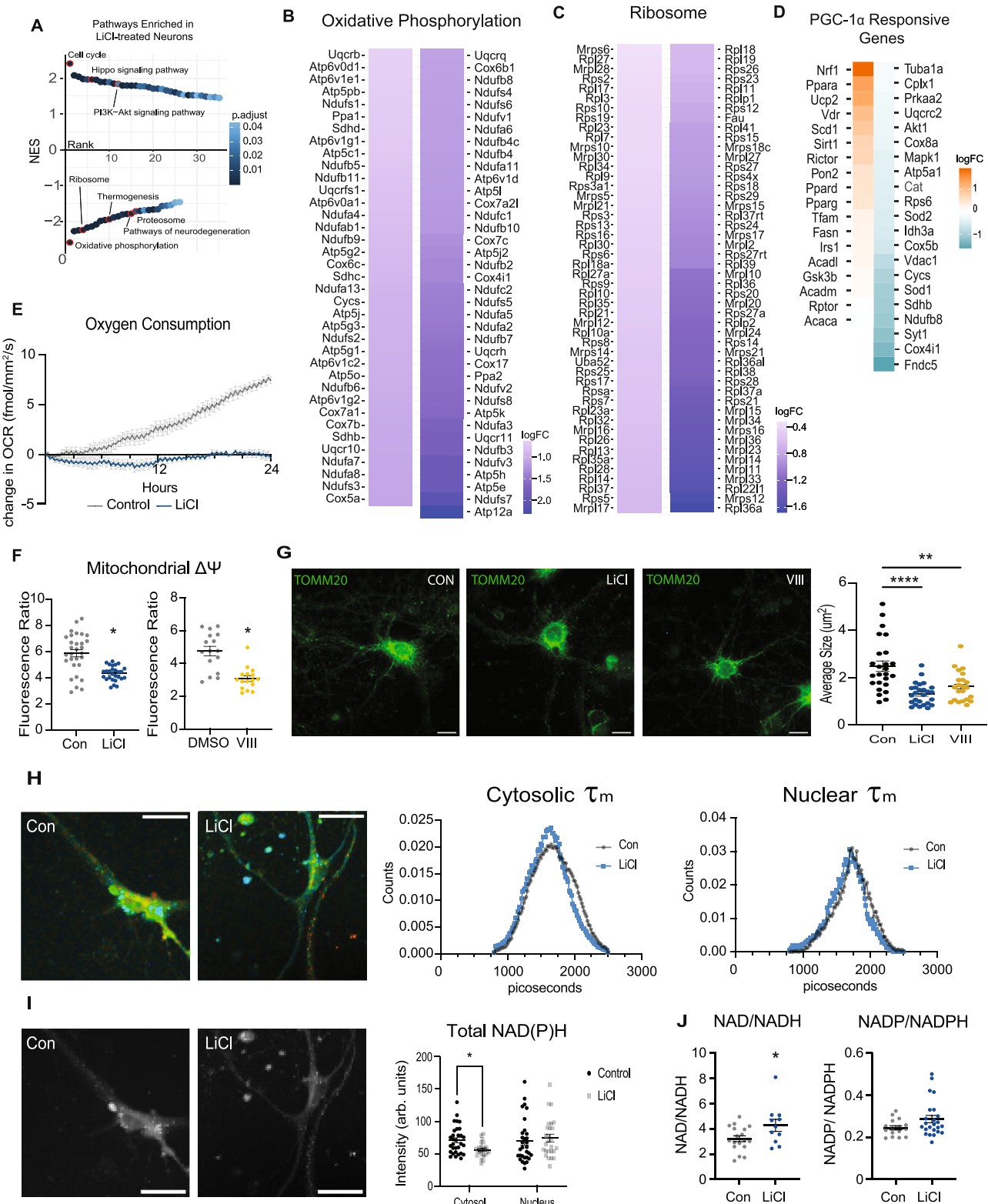

were lower in abundance in $Li_2CO_3$-treated brains. Other genes of interest are related to p53, CREB, and MAPK pathways. Genes with higher expression included Hif3a, Plin4, and Xdh, all connected to metabolic pathways.

Of the 53 overlapping pathways identified as being responsive to $Li_2CO_3$ treatment, the most highly represented signature related to immune and inflammatory pathways (34%), with similar representation from growth and metabolism pathways (46% combined) (Fig. 7D). Two

of the top 5 pathways (Amphetamine, Osteoclast) are largely explained by the presence of Fos, Jun, and Arc. Notch signaling, a key factor in juxtacrine communication, was ranked second among the $Li_2CO_3$ response pathways and included Notch4 and the notch pathway-associated transcription factor effector Maml3 that were increased in abundance, and Hes repressor proteins that were decreased in abundance (Supplementary Data 7). The osteoclast differentiation pathway was also enriched and included Zbtb16 transcriptional repressor linked

**Fig. 5 | Neuronal metabolic response to GSK3β inhibition with LiCl. A** Rank order plot of enriched pathways detected by GSEA of RNA-sequencing of neurons treated for 24 h with 15 mM LiCl ($n = 4$ replicates). Pathways are adjusted $p$-value determined by FDR. **B, C** Heatmap of the significantly changing genes in the oxidative phosphorylation (**B**) and ribosome (**C**) pathways enriched in LiCl-treated neurons. **D** Heatmap of PGC-1α responsive genes detected in the LiCl-treated neuron RNA-seq data. **E** Oxygen consumption of primary neurons treated with LiCl or a control media change measured by RESIPHER monitor for 24 h ($n = 6$ replicates). **F** Mitochondrial membrane potential assessed by JC-1 assay in primary neurons treated with LiCl or GSK3β inhibitor (Inhibitor VIII) for 24 h. **G** Representative images and quantification of mitochondria detected with TOMM20 antibody after 24 h treatment with LiCl or Inhibitor VIII. Mitochondrial size was determined through particle size detection in ImageJ ($n = 25$ control, $n = 26$ LiCl, $n = 22$ inhibitor VIII). **H** Representative images (left) and distributions (right) of NAD(P)H mean fluorescence lifetime of control or LiCl-treated primary neurons. **I** Representative images (left) and quantitation (right) of NAD(P)H fluorescence intensity of control or LiCl-treated primary neurons ($n = 32$ control neurons, $n = 28$ LiCl neurons). **J** NAD/NADH and NADP/NADPH ratios determined by biochemical assay (NAD/NADH: $n = 18$ control, $n = 11$ LiCl; NADP/NADPH: $n = 16$ control, $n = 24$ LiCl). Data shown with error bars noting mean ± SEM (**F, G, I, J**). Asterisks indicate $p$-value < 0.05 by multiple two-tailed $t$-tests (**E**), two-tailed unpaired student's $t$-test (**F, I, J**), or Brown-Forsythe and Welch ANOVA with Dunnett's test (**G**). See Source data file for exact $p$-values. See methods for details on biological replicates for western blots.

to HDACs and Rpskb2 that were increased in abundance, and Dusp6, a negative regulator of MAPK signaling, and Rara (retinoic acid receptor) involved in transcriptional regulation that was lower in abundance. Analyzing each dose individually, the relative enrichment of pathways in cortex from mice on each of the 1.2 g/kg (Fig. 7E), 1.8 g/kg (Fig. 7F), and 2.4 g/kg (Fig. 7G) doses was not equivalent (Fig. 7E). At higher doses of $Li_2CO_3$ pathways linked to insulin signaling are identified including PI3K-Akt signaling (1.8 g/kg) and FoxO signaling (2.4 g/kg). The MAPK growth signaling (Fig. 7H) and circadian rhythm (Fig. 7I) pathways stand out as shared features when the data were analyzed in individual sets, and in general, $Li_2CO_3$ downregulated the expression of genes in both pathways in a dose-dependent manner. A comparison of aging and lithium induced cortical gene expression changes at the pathway level shows considerable overlap (Fig. 7J). These data show that the transcriptional signature detected in primary cell culture is partially conserved at the tissue level, while there are some distinct differences showing that bulk tissue RNA-seq may mask the transcriptional response of specific cell types. Of the 60 pathways enriched between 10 and 30 months of age, 33% were shared in the all-dose lithium pathway enrichment analysis (24% of lithium-responsive pathways). Most of the shared pathways involved immune and inflammatory processes, where the effect of lithium opposed age-related changes.

The R package function scanBam (Rsamtools) was used to extract aligned reads from the Pparg1 genomic region in the RNASeq BAM files. Expression of PGC-1α transcript isoforms did not follow a straightforward linear response to increasing lithium dose. Although not statistically significant, the predicted repression of B1E2 was observed at the 1.2 mg/g dose of lithium carbonate along with activation of the PGC-1α1 and PGC-1α4 (Fig. S10). A panel of 170 PGC-1α-associated genes identified in various independent studies from our lab and others was interrogated to assess PGC-1α activity. While it was clear that there was a response to lithium for many of the PGC-1α-associated genes, it was not all of the genes, and it was not uniform across the lithium doses (Fig. S10). These data indicate that PGC-1α expression and expression of its gene targets are sensitive to lithium in vivo; however, the specifics of gene activation are dependent on the dose of lithium provided.

## Discussion

Brain aging has long been associated with changes in inflammatory and neuronal functional pathways[36], with links to metabolic changes reported more recently[8,37]. Conditions associated with brain aging include cognitive decline and a greater risk for neurodegenerative disease, both of which have been linked to disruption in mitochondrial processes[38,39]. In this study, transcriptional profiles of the cortex from male mice confirm the immune/inflammation and neuronal structural and communication pathways as the most responsive to age and reveal an expansive network of age-sensitive metabolic genes.

Numerous studies now indicate that changes in the transcriptome do not reflect changes in the proteome[26]. There are profound differences in the range and sensitivity of detection between these two platforms, but another consideration is the importance of translational and post-translational regulation. Proteomic analysis confirms age-related changes in metabolism and processes vital for neuronal function but further shows changes in ribosomal and RNA splicing pathways. It is unclear how changes in ribosomal pathways ultimately impact translation, including which transcripts are preferred and how selectivity might be altered. Changes in RNA processing pathways have been reported as a function of age and have recently been proposed as a potential hallmark of aging[40], although the significance of these alternations in terms of function remains unknown at this stage. The impact of age on gene expression of components of the electron transport system are similarly difficult to interpret. Functionally, there is a decline in mitochondrial activity with age[15], at least in the hippocampus; however, we report enrichment of the oxidative phosphorylation pathway at the transcript and protein level in the cortex. It is possible that this reflects a change in the composition of the comprising subunits rather than biogenesis per se. The differences we report are modest and are not ubiquitous across all of the proteins involved in the multi-complex. The most conservative interpretation is that mitochondria are adapting to age while acknowledging that the functional consequence of these changes has yet to be fully understood.

A role for mitochondria in aging is generally accepted, but as yet the role of PGC-1α in aging is not well defined[6,41]. Issues of tissue-type and cell-type specificity in its expression, its interacting partners, and its target genes present a considerable challenge[6]. In primary neurons, three different PGC-1α promoters are active but responsive to different regulatory inputs. All three isoforms of PGC-1α are sensitive to changes in GSK3β activity, and CREB is important at all three promoters; however, the specifics of the influence of regulatory inputs from GSK3β, AMPK, and CREB are distinct at each promoter. Prior studies have reported the induction of PGC-1α1 and PGC-1α4 in response to lithium for cells that do not express the brain-specific promotor-driven isoform. There, the changes in mitochondrial energetics are in the predicted direction. Indeed, lithium also induces these isoforms in primary neurons. The critical distinguishing feature in neuronal metabolic regulation is that metabolic status aligns with expression from the brain-specific promoter, indicating a functional dominance of this isoform. In mice, PGC-1α was previously linked to arborization in neurons of the hippocampus[42] and maintenance of the neuromuscular junction in skeletal muscle[43], possibly secondary to its effects on mitochondria. Direct activities of PGC-1α include regulation of splicing[44] and nuclear to cytosolic mRNA transport[45], each involving a mechanism that requires its RNA binding domain, present on PGC-1α B1E2 and PGC-1α1 but absent on PGC-1α4. Neither of these activities has been specifically linked to PGC-1α B1E2, and the contribution of these regulatory mechanisms to neuronal function is yet unknown.

GSK3β participates in various growth regulatory pathways, including Wnt, insulin/IGF, MAPK, and mTOR signaling[46]. In mouse brains, GSK3β protein abundance increases with age, and one of the unexpected outcomes of this study was the impact of GSK3β inhibition on primary neurons and astrocytes. The repression of PGC-1α at the

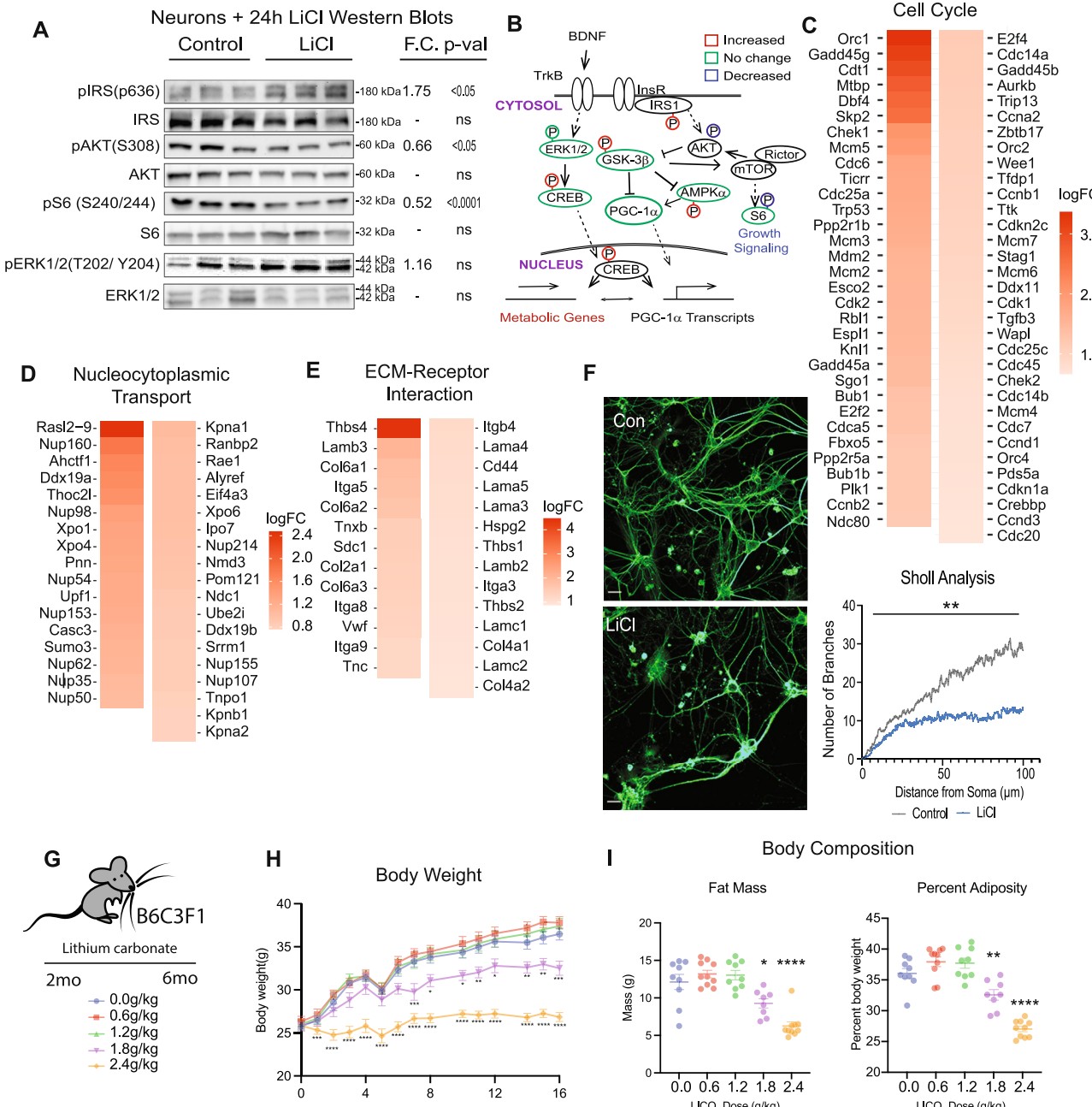

**Fig. 6 | GSK3β plays an integrating role in growth regulation.**
**A** Immunodetection of phosphorylated and total protein levels of IRS, AKT, S6, and ERK1/2 in primary neurons ($n = 6$–8). **B** Schematic of protein expression and phosphorylation in LiCl-treated neurons by Western blot (Figs. 4C and 5A). **C**–**E** Heatmaps of cell cycle, nucleocytoplasmic transport, and extracellular matrix-receptor interaction pathways detected GSEA in lithium-treated neuron RNA-sequencing. **F** Representative images (left) and quantitation (right) of immunodetection of tubulin in control and LiCl-treated primary neurons ($n = 13$ control, $n = 15$ LiCl). Quantitation by Sholl analysis using the Sholl Analysis ImageJ. **G** Schematic of feeding timeline for lithium carbonate mouse study. Mice were fed lithium

carbonate-containing food at 0, 0.6, 1.2, 1.8, and 2.4 g/kg daily for 16 weeks. **H** Body weights of the mice fed diets containing the five doses of $LiCO_3$ ($n = 10$ mice/ diet). **I** Body composition analysis for the $LiCO_3$-fed mice. Total fat mass (left) and fat mass as a percent of total adiposity (right). 0.0: 9 mice, 0.6: 10 mice, 1.2: 9 mice, 1.8: 8 mice, 2.4: 10 mice. Data shown with error bars noting mean ± SEM. Statistical significance was determined by two-tailed unpaired Student's $t$-test (**A**), two-tailed unpaired student's $t$-test with Welch's correction (**F**) or ordinary one-way ANOVA with Dunnett's test (**H**, **I**). See Source data file for exact $p$-values. See methods for details on biological replicates used for western blots.

brain promoter in response to LiCl or GSK3β inhibitor VIII is opposite to what is observed at the canonical and alternate promoters in neurons and in other cell types where the brain promoter is not active. This raises the possibility that PGC-1α is part of a broader molecular strategy that allows different brain-resident cells to independently modulate cellular energetics in response to extracellular stimuli. The importance of GSK3β as a regulatory effector in central metabolism

and growth signaling is underscored by the primary neuron and in vivo brain response to LiCl (neurons) and $Li_2CO_3$ (mice). In the cortex, the changes in PGC-1α isoforms were not a straightforward linear response to increasing lithium dose and are likely to be highly influenced by the fact that (a) multiple cell types are represented in the RNASeq data and (b) that cells act as communities and engage in cross-talk in their response to a given stimulus, a feature that cannot be captured in

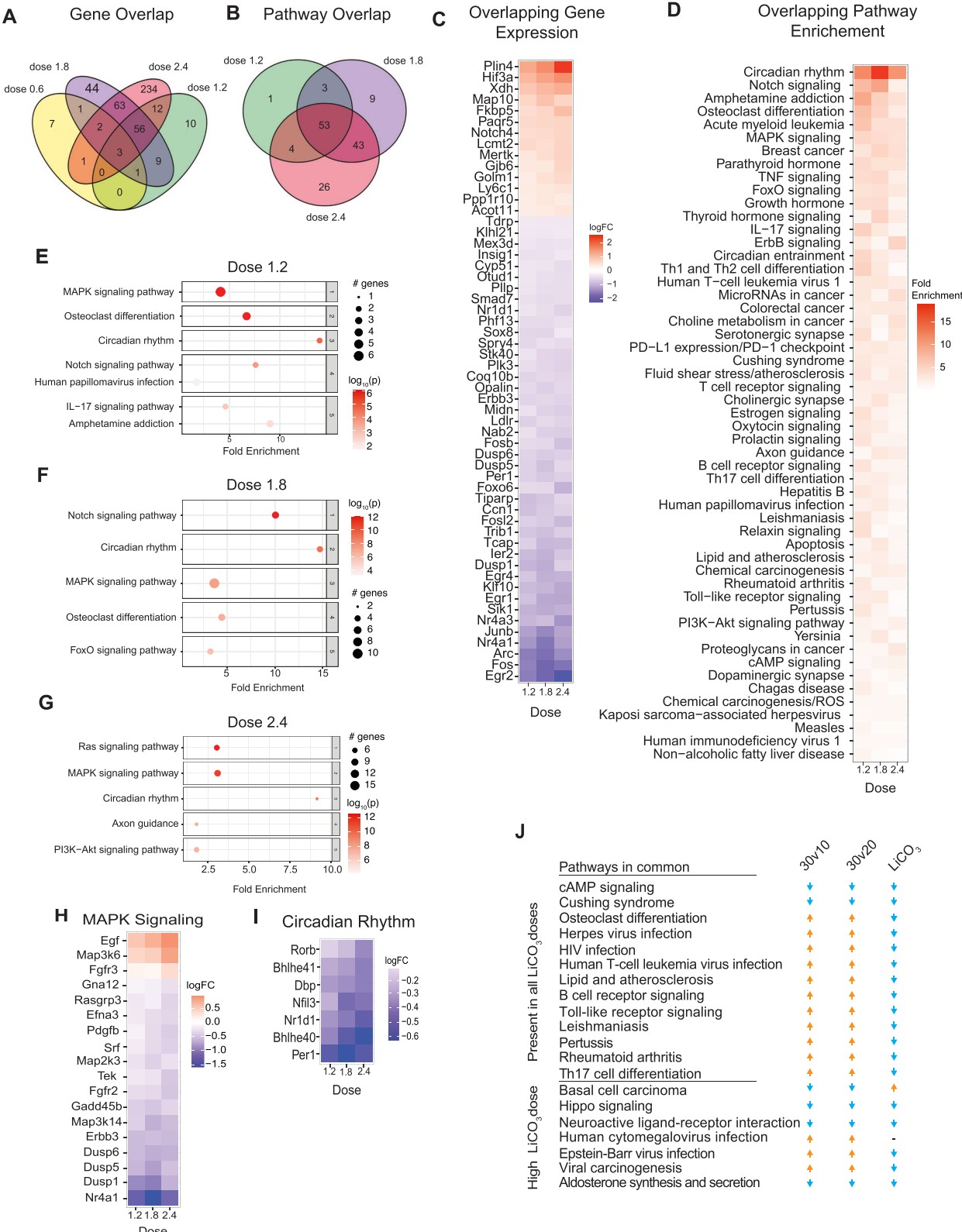

**Fig. 7 | LiCl impacts brain growth and metabolism pathways. A** Venn diagram of significant genes detected in the brains of mice treated with 4 different doses of LiCO$_3$ compared to control mice ($n = 4$ mice/ diet). **B** Venn diagram of pathways enriched in the brains of mice given 1.2, 1.8, and 2.4 g/kg/day. Pathways were detected by PathfindR. **C** Heatmap of the 56 genes that are differentially expressed in all 3 doses. **D** Heatmap of 53 overlapping enriched pathways detected by PathfindR. **E–G** Dotplots of top 5 enriched pathways in the 1.2 (**E**), 1.8 (**F**) and 2.4 (**G**) g/kg mouse brains. Significant pathways were determined by an FDR < 0.05. **H, I** Heatmap of the significantly changing genes in MAPK Signaling (**H**) and Circadian Rhythm (**I**) pathways across each of 3 doses of LiCO$_3$. **J** Table displaying common KEGG pathways between the 30v10 30v20 and the LiCO$_3$ diets comparison. An up arrow indicates positively enriched in 30 months compared to 10 or 20 months, respectively, or in LiCO$_3$ diets compared to the control diet.

isolated primary cells. It will be important to dissect how the neuronal PGC-1α response to GSK3β might be influenced by juxtacrine effects, particularly whether contributions from astrocytes and microglia might modify the response reported here for neurons in primary culture.

The regulation of GSK3β by phosphorylation at serine 9 disables its ability to phosphorylate targets that have a pre-primed phosphorylation directly upstream from the consensus site[47]. The phosphorylation at S9 blocks the opening that "recognizes" the primed site on GSK3β targets. We previously reported increased GSK3β phosphorylation is increased in the hippocampus of these same mice subsequent to lithium treatment[14]. Furthermore, lithium and Inhibitor VIII sensitive direct phosphorylation of PGC-1 in context of the oxidative stress response impacts the stability of the transcriptional coactivator[29]. A similar mechanism has been reported for the circadian regulators BMAL and Rev-Erb-alpha, although in those cases, there are opposing outcomes of GSK3β-induced phosphorylation with increased and decreased stability, respectively[48,49]. There are multiple putative GSK3β sites on PGC-1α, but we do not know the functional consequences of site-specific phosphorylation and/or if a suite of kinases may be acting in parallel. A newer regulatory site identified on GSK3β involves acetylation at K15[50], a modification is predicted to stabilize it, but its physiological significance and how it might impact kinase activity is unclear. The direct consequence of changes in levels of GSK3β protein on GSK3β activity and/or target selectivity in aged tissues will need to be more fully explored.

A primary limitation of this study is that both in vivo studies, brain aging and the response to $Li_2CO_3$, were conducted in male mice only. Growing evidence points to profound sex dimorphism at the cellular and molecular level[51]. Even while the physiological traits of aging manifest quite similarly for both sexes, the key pathways and processes contributing to aging are quite sex dimorphic. For example, in rats, age-related changes in brain inflammatory cytokines have been reported for both males and females but appear to be more pronounced in females compared to males[52]. It will be important to uncover shared and sex-specific effects of aging on the brain and whether the interventions geared toward delaying aging might need to be optimized for sex. Despite this limitation, it seems clear that there are brain-specific mechanisms for controlling energetics and that changes in mitochondrial pathways are at least coincident with changes in inflammatory and neuronal functional pathways with age. Differences in how and when PGC-1α is activated are likely to play a role in establishing cell-type metabolic specificity, and differences in the PGC-1α response would allow for intracellular coordination in metabolic adaptation, where a single stimulus evokes distinct outcomes according to cell type. Given the breadth of pathways and processes influenced by mitochondrial status, there is considerable interest in advancing what we know about the contribution of mitochondria to disease and motivation to go beyond the concept of "dysfunction"[3]. Data from this study supports the idea that mitochondrial processes could be targeted to delay age-related changes in the brain and even potentially applied as a preventative measure for neurodegenerative disease.

## Methods
### Experimental model and subject details
**Aging cohort.** Six-week-old male B6C3F1 hybrid mice were obtained from Harlan Laboratories (Madison, WI, USA) and housed under controlled pathogen-free conditions in accordance with the recommendations of the University of Wisconsin Institutional Animal Care and Use Committee. Mice were fed 87 kcal week$^{-1}$ of the control diet (Bio-Serv diet #F05312) from 2 months of age and were individually housed. This level of calorie intake is ~95% of ad libitum for the B6C3F1 strain, so all food was consumed. By 30 months of age, the mortality of the control animals was ~45%, consistent with the expected lifespan for this strain. Mice were euthanized by cervical dislocation at 10, 20, and 30 months of age.

**Lithium diet study.** Six-week-old male B6C3F1 hybrid were obtained from Harlan Laboratories (Madison, WI, USA) and housed under controlled pathogen-free conditions in accordance with the recommendations of the University of Wisconsin Institutional Animal Care and Use Committee. Mice were fed 87 kcal week$^{-1}$ of the control diet (F05312; Bio-Serv, Flemington, NJ, USA) and were individually housed with ad libitum access to water. This level of food intake is ~95% ad libitum for the B6C3F1 strain, so all food was consumed. Following two weeks of facility acclimation, mice were randomized into five treatment groups fed the control diet supplemented with increasing concentrations of dietary lithium carbonate (2 months old; $n = 10$/group): Group 1) 0.0 g/kg/day $Li_2CO_3$;Group 2) 0.6 g/kg/day $Li_2CO_3$; Group 3) 1.2 g/kg/day $Li_2CO_3$; Group 4) 1.8 g/kg/day $Li_2CO_3$; Group 5) 2.4 g/kg/day $Li_2CO_3$. $Li_2CO_3$-supplemented mice were administered an additional drinking bottle containing saline (0.45% NaCl) to offset polyuria, a common side effect of lithium treatment. Mice consumed dietary lithium for 4 months and were euthanized at 6 months of age. Mice were weighed every 1 and 2 weeks throughout the duration of the study. Body composition analysis was conducted using the Lunar PIXImus machine and software.

Brains were isolated, bisected, embedded in OCT, frozen in liquid nitrogen, and stored at −80 °C until further processing.

**Primary cell culture studies.** Wild-type male and female C57BL6J mice were obtained from Jackson Laboratories (Bar Harbor, ME) and housed under controlled pathogen-free conditions in accordance with the recommendations of the University of Wisconsin Animal Care and Use Committee. Mice were allocated to breeding pairs at 8 weeks of age with two animals per cage.

All animal protocols were approved by the University of Wisconsin Institutional Animal Care and Use Committee (MV1774−aging; MV2579−Lithium diet) or the Institutional Animal Care and Use Committee at the William S. Middleton Memorial Veterans Hospital (RA0006−primary cell isolation).

### Method details
**RNA sequencing.** RNA extraction was completed using a Direct-zol RNA kit (Zymo Research, Irvine, CA) according to the manufacturer's instructions. Each RNA library was generated following the Illumina TruSeq RNA Sample Preparation Guide and the Illumina TruSeq RNA Sample Preparation. Purified total RNA was used to generate mRNA libraries using NEBNext Poly(A) mRNA Magnetic Isolation Module and NEBNext Ultra RNA Library Prep kit for Illumina (Illumina Inc., San Diego, CA, USA). Quality and quantity were assessed using an Agilent DNA1000 series chip assay and Invitrogen Qubit HS Kit (Invitrogen), respectively. Sequencing reads were trimmed to remove sequencing adaptors and low-quality bases[53], aligned to mm10 reference genome using the STAR aligner, and alignments used as input to RSEM for quantification. Differential gene expression analysis was performed via EdgeR generalized linear model (GLM) method. GSEA analysis was performed via gseKEGG package in clusterProfiler. P value cutoff was set to 0.05, and pAdjustMethod was fdr. Pathway analysis for the $LiCO_3$ mouse brain dataset was conducted using PathfindR. P value threshold was set to 0.05 and adjustment method was FDR.

### Proteomics
**Enzymatic "In Liquid" Digestion.** Total protein extracted from mouse brain punches was re-solubilized and denatured in 8 M Urea/100 mM Tris-HCl (pH8.3)/100 mM NaCl, concentration quantitated from BCA and 50ug taken for subsequent digestion and label-free nanoLC-MS/MS profiling as follows. Total volume of each sample was adjusted to

50 μl with 8 M Urea and subsequently diluted to 180 μl for reduction step with: 7.5 μl of 25 mM DTT, 15 μl MeOH, 15 μl 500 mM NH$_4$HCO$_3$ (pH8.5) and 92.5 μl of 25 mM NH$_4$HCO$_3$ (pH8.5). Samples were incubated at 52 °C for 15 min, cooled on ice to room temperature then 9 μl of 55 mM chloroacetamide was added for alkylation and incubated in darkness at room temperature for 15 min. The reaction was quenched by adding 24 μl of 25 mM DTT. Subsequently 8 μl of Trypsin/LysC solution [100 ng/μl of 1:1 Trypsin (Promega) and LysC (FujiFilm) mix in 25 mM NH$_4$HCO$_3$] and 79 μl of 25 mM NH$_4$HCO$_3$ (pH8.5) were added and brought to 300 μl final volume. Digestion was conducted for 2 h at 42 °C then additional 4 μl of trypsin/LysC solution added and digestion proceeded o/n at 37 °C. The reaction was terminated by acidification with 2.5% TFA [Trifluoroacetic Acid] to 0.3% final.

**NanoLC-MS/MS.** Digests were cleaned up using OMIX C18 SPE cartridges (Agilent, Palo Alto, CA) per manufacturer protocol and eluted in 20 μl of 70/30/0.1% ACN/H$_2$O/TFA, dried to completion in the speedvac and finally reconstituted in 40 μl of 0.1% formic acid. Peptides were analyzed by nanoLC-MS/MS using the Agilent 1100 nanoflow system (Agilent) connected to hybrid linear ion trap-orbitrap mass spectrometer (LTQ-Orbitrap Elite™, Thermo Fisher Scientific) equipped with an EASY-Spray™ electrospray source. Chromatography of peptides prior to mass spectral analysis was accomplished using capillary emitter column (PepMap® C18, 3 μM, 100 Å, 150 × 0.075 mm, Thermo Fisher Scientific) onto which 1.5 μl (~1.9ug) of extracted peptides was automatically loaded. NanoHPLC system delivered solvents A: 0.1% (v/v) formic acid, and B: 99.9% (v/v) acetonitrile, 0.1% (v/v) formic acid at 0.50 μL/min to load the peptides (over a 30 min period) and 0.3 μl/min to elute peptides directly into the nano-electrospray with gradual gradient from 0% (v/v) B to 30% (v/v) B over 155 min and concluded with 10 min fast gradient from 30% (v/v) B to 50% (v/v) B at which time a 7 min flash-out from 50–95% (v/v) B took place. As peptides eluted from the HPLC-column/electrospray source survey MS scans were acquired in the Orbitrap with a resolution of 120,000 followed by CID-type MS/MS fragmentation of 20 most intense peptides detected in the MS1 scan from 350 to 1800 m/z; redundancy was limited by dynamic exclusion.

**Data analysis.** Elite acquired MS/MS data files were searched using Proteome Discoverer (ver. 2.5.0.400) Sequest HT search engine against Uniprot *Mus musculus* database (UP000000589, 06/16/2020 download, 63,686 total sequences) along with a cRAP common lab contaminant database (116 total entries). Static cysteine carbamidomethylation, and variable methionine oxidation plus asparagine and glutamine deamidation, 2 tryptic miss-cleavages and peptide mass tolerances set at 15 ppm with fragment mass at 0.6 Da were selected. Peptide and protein identifications were accepted under strict 1% FDR cut offs, at least two peptide spectrum matches per protein and with high confidence XCorr thresholds of 1.9 for $z = 2$ and 2.3 for $z = 3$. Strict principles of parsimony were applied for protein grouping. Chromatograms were aligned for feature mapping and ion intensities were used for precursor ion quantification using unique and razor peptides. Normalization was performed on total peptide amount and scaling on all average. Protein abundance was determined from summed peptide abundances and ANOVA (individual proteins) was executed for hypothesis testing.

## Primary cultures

**Primary neurons.** Primary cortical neurons were isolated from brains of P0 mice. Briefly, the cortices were harvested from pups into ice-cold HBSS where the midbrain and meninges were removed. Cortices were pooled in groups of 2. Brain tissue was minced and digested in 0.25% trypsin for 20 min at 37 °C. Trypsin was quenched with DMEM/10%FBS/1%Pen/Strep, and cells were dissociated and counted prior to plating on poly-d-lysine coated plates. Cultures were then fully changed to

Neurobasal Plus Media (2% B27$^+$, 1% GlutaMax, 1% Pen/Strep) the subsequent day. Upon first change to Neurobasal media, neurons were treated with 1 μM AraC (Cytosine β-D-arabinofuranoside hydrochloride) to eliminate glial cell populations. Media was changed by ½ volume every 3 and 4 days until experiments. All primary neuron experiments were conducted between day 10–14 in vitro.

**Primary astrocytes.** Primary cortical astrocytes were dissociated similarly to neurons and plated in DMEM/10% FBS/1% Pen/Strep. Briefly, cortices were collected from P1-2 mouse pups and pooled in groups of 2. Cells were dissociated enzymatically in 0.25% Trypsin for 20 min. Cells were further dissociated by trituration. Cells from 2 cortices were plated on a T-75 cell culture flask and grown in DMEM supplemented with 10% FBS and 1% Pen/Strep. Astrocytes proliferated to confluency and were subsequently treated with 10 μM AraC for 48 h to eliminate non-astrocyte glial populations. Upon microglial removal, astrocytes were then passaged to a T-175 to ensure astrocytic growth. Cells were passaged for experiments at ~80% confluence and plated for a one-day growth.

**Neural stem cells (NSCs).** Hippocampi were dissected in cold HBSS and dissociated using the GentleMACS Dissociator (Miltenyi Biotec) and MACS Neural Tissue Papain Dissociation Kit (Miltenyi Biotec) according to the manufacturer's protocol. NSCs were cultured in DMEM/F12 media (Invitrogen) supplemented with GlutaMAX, FGF (20 ng/mL), EGF (20 ng/mL) and Heparain (5 μg/mL) for the indicated number of days prior to assay.

All cells were maintained at 37 °C at 5% CO$_2$. All cortical cultures were produced through pooling tissue in groups of two. Therefore, we cannot report the sex of the cells in the cultures.

**Drug treatments.** All drugs were administered for 24 h through a 50% media change. Drugs were administered at final concentrations as follows: Lithium chloride (Sigma Aldrich; L7026), 15 mM. Actinomycin D (Fisher Scientific AC294940010), 8 μM. Inhibitor VIII (Millipore Sigma 361549), 15 μM. Compound C (Millipore Sigma; 171260), 8 μM. ANA-12, 20 μM. CREB inhibitor 666-15 (Millipore Sigma 5383410001), 1 μM.

## Histochemistry
Serial cryostat sections of 10 μM in thickness were cut at −20 °C with a Leica Cryostat (Fisher Supply, Waltham, MA, USA), defrosted and air-dried, and stained for GSK3β (9315; Cell Signaling Technologies, Danvers, MA, USA) as previously described[54]. Images were acquired on a Leica DM4000B microscope equipped with an HCX PL FLUOTAR 20×/0.50 objective. Photographs were taken with a Retiga 4000 R digital camera (QImaging Systems, Surrey, BC, Canada). Camera settings were optimized for each stain, and for uniformity, all images were taken with identical settings, fixed light levels, and fixed shutter speeds.

## RT-qPCR
Cells were treated as outlined above and RNA was collected 24 h after treatment. For NSC differentiation and primary neuron maturation, cells were lysed at day in vitro indicated above. Cells were lysed with Trizol and RNA was isolated using Zymo Research Direct-zol RNA MiniPrep Kit. RT-qPCR was conducted using iTaq Universal SYBR Green Supermix (1725121, Bio-Rad, Hercules, CA, 94547). Primer sequences for all transcripts can be found in Supplementary Data 5.

## Read density of Pparc-1a locus determination
Fastq files of raw sequencing reads from GSE52564[55] were downloaded using SRA Toolkit (https://github.com/ncbi/sra-tools/wiki/). Sequences were aligned to the mouse RefSeq assembly (GRCm39) using Rsubread[56]. Aligned reads within the Ppargc1a locus (chr5:51,611,591-52,273,171) were extracted using scanBam() from Rsamtools[57]. Read density was plotted using ggplot2[58].

## Multiphoton laser scanning microscopy

Conducted as previously described. Briefly, cells were grown on glass coverslips, fixed in 10% formalin, and mounted using Fluoromount (Thermo Scientific). Images were captured using a Nikon 60 × 1.3apo objective and a 457/40 filter. The acquisition time for neuron-astrocyte comparison was 120 s. Lithium-treated neuron experiments were conducted using a 60-s acquisition time. Image analysis was conducted using SPCImage 8.0 (https://www.becker-hickl.com/products/spcimage/).

## Promoter region transcription factor binding site determination

Predicted CREB binding sequences present in the promoter regions of the *Ppargc-1a* gene were predicted using transcription factor binding motif prediction software (http://tfbind.hgc.jp/). The prediction was made using reported CREB-ChIP sequencing peaks and the strength of CREB prediction was based on the density score produced by the number of potential CREB binding sites[59]. Confidence values and density scores were then z-scored to determine the predicted binding sites.

## Western Blot, and Immunofluorescence

Cells were lysed, and protein was extracted in a modified RIPA buffer containing protease and phosphatase inhibitors (P8340 and 524624, respectively; Sigma-Aldrich, St. Louis, MO, USA). Proteins were detected by immunoblotting using standard techniques. Antibodies used were all acquired from Cell Signaling Technologies unless otherwise noted and used at the manufacturer's recommended dilution. Antibodies used were: pGSK3β(S9) (#9336), GSK3β (9315S), pCREB (S133) (ab32096; Abcam), CREB (#9104), pAMPKα (T172) (#2535S), AMPKα (#2532S), pIRS-1 (S636) (#2388S), IRS-1 (#2382S), pAKT (T308) (#13038S), AKT (#4691), pS6 (S240/244) (#2215S), S6 (#2217S), pERK1/2 (T202/ Y204) (#4370S), ERK1/2 (#4695S). Western blot densitometry was conducted in Adobe Photoshop to detect band intensity.

Immunofluorescence images were acquired on a Leica DM4000B microscope equipped with a Leica N Plan 40×/0.65 objective. Photographs were taken with a Retiga 4000 R digital camera (QImaging Systems, Surrey, BC, Canada). Camera settings were optimized for each stain, and for uniformity, all images were taken with identical settings, fixed light levels, and fixed shutter speeds. Cells were grown on glass coverslips coated with poly-d-lysine. Cells were fixed in 10% formalin for 10 min, permeabilized with 0.3% Triton X-100 in PBS (PBS-T) for 1 h, and incubated with primary antibody diluted in 1% BSA overnight. Primary antibodies used were anti-a-Tubulin primary antibody (1:200) (Sigma-Aldrich #T6199) and Tomm20 (1:84.2) (Abcam ab56783). Cells were washed with PBS and then incubated with a secondary antibody, Goat anti-mouse IgG,AlexaFluor 488, for 1 h at room temperature. Coverslips were mounted using Fluoromount (Thermo Scientific). Sholl analysis was conducted with ImageJ (NIH, Wayne Rasband, http://rsb.info.nih.gov/ij/). using the Sholl Analysis plugin (https://imagej.net/plugins/sholl-analysis).

## Bioassays

**Oxygen consumption (OC).** OC was measured using a Resipher oxygen consumption monitor. OC was measured for 24 h prior to treatment to establish basal respiration. Change in OC was determined by comparing the rate at 15-min intervals taken for 24 h post-treatment.

**JC-1 Assay.** Mitochondrial membrane potential was quantified with JC-1 dye (Invitrogen, Waltham, MA). Neurons were treated with 15 mM LiCl for 24 h and then incubated with 1 μg/mL JC-1 dye for 30 min. Cells were washed with PBS prior to fluorescence detection. Fluorescence was measured using excitation/emission wavelengths of 535/590 nm and 485/530 nm.

**NAD(P) and NAD(P)H.** NAD(P) and NAD(P)H concentrations were quantified with NAD(P)- and NAD(P)H-Glo assay kits (Promega) according to manufacturer instructions. Neurons were treated with 15 mM LiCl for 24 h prior to assay.

**Cell viability.** Cell viability was measured using CyQuant assay (ThermoFisher C7026) following manufacturer's instructions. Neurons were grown in black-walled 96 well plates. Neurons were treated with 15 mM LiCl for 24 h prior to assay.

## Statistics and reproducibility

All data capture experiments were conducted in samples derived from a minimum of four biological replicates unless otherwise noted (each n is a biological replicate/ independent data point). Each image-based data capture experiment involved at least five individual cells per pooled cortex sample per treatment. Because these are primary neurons and not immortalized cell lines, we employed a strategy where each cell was considered individually for data capture and subsequent analysis. For all RT-qPCR and western blot experiments, a replicate is defined as one culture dish derived from two pooled cortices All replicates shown are biological and not technical. Outliers were identified by ROUT using a threshold of $Q = 0.05$. All student's t-tests were two-tailed. Unless otherwise noted, datasets involving two groups were analyzed by unpaired Student's t-test, datasets with three groups, ordinary one-way ANOVA was used, and datasets with four groups, two-way ANOVA with Tukey's Multiple comparisons test was used. Unless otherwise noted, error bars represent mean +/− standard error. Statistical analysis was conducted using GraphPad Prism and R studio. RNA-sequencing and proteomics were analyzed using R. Asterisks on figures denote significance levels: * $p < 0.05$, ** $p < 0.01$, *** $p < 0.001$, **** $p < 0.0001$.

## Reporting summary

Further information on research design is available in the Nature Portfolio Reporting Summary linked to this article.

## Data availability

All RNA sequencing data from this study are deposited in the GEO repository (GSE246478). The mass spectrometry proteomics data have been deposited to the ProteomeXchange Consortium via the PRIDE partner repository with the dataset identifier PXD060832 and 10.6019/PXD060832. Cell type-specific read density along the Ppargc1a locus was quantified using publically available RNA-seq data (GSE52564). Source data are provided as a source data file. Uncropped images of immunoblots can be found in the supplemental information (Fig. S11). All other data is available upon request. Further information and requests for resources and reagents should be directed to Lead Contact Dr. Rozalyn Anderson (rozalyn.anderson@wisc.edu). Source data are provided with this paper.

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

## Acknowledgements

This work was supported by NIH AG067330 (R.M.A.), NIH AG057408 (R.M.A., L.P.), NIH CA268069 (K.W.E.), the Simons Foundation (R.M.A.), and NIH training fellowships AG000213 (D.C.S.) and DK007665 (E.R.M.). This study was conducted using resources and facilities at the William S. Middleton Memorial Veterans Hospital, Madison, WI. The authors would like to thank Jenu Chacko for his technical assistance in conducting the fluorescence lifetime imaging experiments.

## Author contributions

D.C.S. and E.R.M. contributed equally to this work. D.C.S., E.R.M., T.W.R., J.P.C., and T.J.P. generated data, conducted analysis, and along with D.L.M., L.P., and K.W.E., all contributed to data interpretation. D.C.S., E.R.M., and R.M.A. wrote the manuscript.

## Competing interests

The authors declare no conflict of interest.
