## [Transparent Peer Review file · Nature Communications]

Neuron-specific isoform of PGC-1 α regulates neuronal metabolism and brain aging

Corresponding Author: Professor Rozalyn Anderson

Version 0:

Reviewer comments:

Reviewer #1

(Remarks to the Author)

The author's responded appropriately to my first two concerns by performing additional experiments and including the results in the revised manuscript. Regarding my third concern, primary neurons can be readily transfected using AAV or lentivirus constructs and many published studies have used this approach to knock down targeted proteins. However, I do understand that the present manuscript is a timely contribution to the field without B1E2 isoform knockdown.

(Remarks on code availability)

Reviewer #3

(Remarks to the Author)

In the revised manuscript, the authors have addressed some conceptual and technical concerns made by this reviewer. Nevertheless, multiple major concerns remain to be addressed, which leaves the proposed role of PGC1 α -GSK3 β axis in brain aging insufficiently supported, especially regarding the differential transcriptional regulation of PGC1 α across cell types. Further, I could not assess any responses to technical or methodological comments as the entire Methods section was missing from the submitted documents.

- In vivo data that support the differential transcription regulation of PGC1 α in neurons vs astrocytes remain unavailable. All data related to PGC1 α B1E2 were either derived from primary cell cultures (and thus the age effect was not determined) or bulk cortical tissues.
- This revision did not address the concern on the conceptual integration of the three parts included in the manuscript. The transcriptomic and proteomic analyses (Figs. 1-2) still provide very limited value to the proposed mechanism involving "differential PGC-1 α isoform expression and differences in GSK3 β sensitivity across cell types". Similarly, in the lithium study, PGC1 α transcriptional regulation was still not investigated, whether at bulk tissue level or at cell type level. The authors added a reference to their previous publication involving lithium treatment (Martin et al, 2018) but in that study PGC1 α transcript variance was not investigated at all.
- While new data on mitochondrial size and membrane potential in neurons treated with GSK3 β inhibitor VIII were added, key information regarding the effect of GSK3 β inhibition on PGC1 α transcriptional profile using a specific GSK3 β inhibitor were still not provided.
- Lithium treatment using young mice provided limited insights to the role of GSK3 β in brain aging, which was still not addressed.
- Even from a brain metabolism perspective (not relevant to aging), data presented in this study provides limited new mechanistic insights comparing to the group's previous publication (Martin et al, 2018) except for the new RNAseq dataset and a list of enriched pathways.
- The authors need to address the discrepancy in the effect of GSK3 β inhibition on neuronal / brain metabolism across two studies from the authors' group. In the present study, lithium was found to reduce neuronal mitochondrial respiration and potential, whereas in Martin et al, 2018, lithium was found to increase mitochondrial respiration and membrane potential.

(Remarks on code availability)

Version 1:

Reviewer comments:

Reviewer #3

(Remarks to the Author)

While most of my concerns have been addressed in the revised manuscript, there is still some room for improvement. Nonetheless, the manuscript meets an acceptable standard in its current form.

NCOMMS-24-27624-T

We are grateful for the opportunity to submit our revised paper. We note that only one reviewer had remaining minor concerns that we address in our responses below, indicating where the changes have been made in the manuscript and figures.

Comment 1: In the revised manuscript, the authors have addressed some conceptual and technical concerns made by this reviewer. Nevertheless, multiple major concerns remain to be addressed, which leaves the proposed role of PGC1 α -GSK3 β axis in brain aging insufficiently supported, especially regarding the differential transcriptional regulation of PGC1 α across cell types. Further, I could not assess any responses to technical or methodological comments as the entire Methods section was missing from the submitted documents.

Response: We address PGC-1 α expression across cell types but concede the point that we did not define the impact of GSK3 β inhibition in other cell types beyond primary neurons. We have revised the abstract to focus on PGC-1 α as a part of a growth and metabolism network relevant to aging.

Section: Abstract

The missing methods section was an error on our part. We are now providing it in revised form with the new experiments.

Section: "Methods" – lines 565 - 782

Comment 2: In vivo data that support the differential transcription regulation of PGC1 α in neurons vs astrocytes remain unavailable. All data related to PGC1 α B1E2 were either derived from primary cell cultures or bulk cortical tissues.

Response: To address this concern, we analyzed publicly available data on cell-type specific transcriptomes generated by the Barres/Wu labs (PMID:25186741). Based on read density across the promoter and coding regions, it is clear that PGC-1 α B1E2 is only expressed in neurons and not in astrocytes. Those data are provided in a new panel (Fig.3D). The differential expression from the alternate and canonical promoter between cell types can be seen in the normalized read counts (Fig.S4). These new data support the original claim that age-related changes in expression of B1E2 (Fig.3E) can reasonably be assigned to the neuronal population. As an aside, repeated attempts to isolate neurons and glia from older mice using the Miltenyi MACS system were sadly unsuccessful. We intend to follow up with colleagues who have generated snRNASeq from mice of different ages once they release those data and have initiated our own studies of aging and cell type specificity in B1E2 expression using RNAScope. In confidence, we can share that in our rhesus monkey snRNASeq/snATACSeq data generated from the prefrontal cortex, we find chromatin accessibility across B1 to B4 to E2 for excitatory and inhibitory neurons and completely closed configuration for microglia and oligodendrocytes. For astrocytes, accessibility at this region of the locus is massively reduced.

Section: "PGC-1 α transcript isoforms in the brain and cell type specificity", paragraph 1, lines 193 – 198.

Comment 3: This revision did not address the concern about the conceptual integration of the three parts included in the manuscript. The transcriptomic and proteomic analyses (Figs. 1-2) still provide very limited value to the proposed mechanism involving "differential PGC-1 α isoform expression and differences in GSK3 β sensitivity across cell types".

Response: We appreciate the reviewer's point but would argue that the purpose of showing the bulk profiling data at the start of the paper is to provide the aging context. The new insight is that changes in pathways previously linked to aging (inflammation and neuronal functional processes; Fig.1) are coincident with metabolic changes identified at the transcriptional level and corroborated at the proteomic level (Fig.2). That observation was the springboard for our investigation into metabolic regulation of brain metabolism, i.e. what might be responsible for age-related changes in metabolism. In the course of that work, it became clear that the differences in metabolic status among cell types

were linked to differences in PGC-1a isoform expression. Our investigation of cell type specificity in metabolic regulation used the known neuromodulator and growth-sensitive kinase GSK3b and showed key differences in PGC-1a promoter response to GSK3b inhibition. The in vivo lithium treatment study shows that age-sensitive pathways in the brain can be harnessed via GSK3b inhibition and that PGC-1a and its gene targets are engaged in that response. Together, these data provide support for the interconnection of growth signaling and metabolic pathways in brain aging and suggest a role for PGC-1a and GSK3b therein.

Comment 4: Similarly, in the lithium study, PGC1a transcriptional regulation was still not investigated, whether at bulk tissue level or at cell type level. The authors added a reference to their previous publication involving lithium treatment (Martin et al, 2018) but in that study PGC1a transcript variance was not investigated at all.

Response: We are happy to have the opportunity to address this oversight. The RNA extractions used for the RNA-Seq transcriptional analysis of the in vivo effect of lithium were used entirely for the library preparation. With no additional material set aside, we were prevented from doing a PCR on RNA extracted from those exact tissue specimens. Fortunately, we were able to use the R package function scanBam (Rsamtools) to extract aligned reads from the Pparg1 genomic region in the RNASeq BAM files. In general, we see the predicted repression of B1E2 at the 1.2mg/g dose of lithium carbonate along with activation of the alpha 1 and alpha 4, but that pattern was not sustained at higher doses of the GSK3b inhibitor. We show this in a new data element (Fig.S10). To dig a little deeper, we surveyed a panel of 170 PGC-1a-associated genes identified in various independent studies from our lab and others. While it was clear that there is a response to lithium for many of the PGC-1a-associated genes, it was not a uniform response across all gene targets, and it was not linear across the lithium doses. These data are shown in a new panel in the supplement (Fig.S10). For now, we limit our interpretation to the following: “PGC-1a expression and expression of gene targets are sensitive to lithium in vivo; however, the specifics of gene activation are dependent on the dose of lithium carbonate provided”.

Section: “Lithium impacts brain growth and metabolism pathways,” paragraph 4, lines 442 – 452. Later, we touch on this point again with the following statement: “In the cortex, the changes in PGC-1a isoforms were not a straightforward linear response to increasing lithium dose and are likely to be highly influenced by the fact that a) multiple cell types are represented in the RNASeq data and b) that cells act as communities and engage in cross-talk in their response to a given stimulus, a feature that cannot be captured in isolated primary cells”.

Section: “Discussion,” paragraph 4, lines 508 – 512.

Comment 5: While new data on mitochondrial size and membrane potential in neurons treated with GSK3β inhibitor VIII were added, key information regarding the effect of GSK3β inhibition on PGC1a transcriptional profile using a specific GSK3β inhibitor were still not provided.

Response: We think there may be some confusion here – we provided data on PGC-1a expression in response to inhibitor VIII treatment alone, where outcomes match the effect of lithium (Fig.4f) and in combination with lithium, where there is no additive effect.

Section: “PGC-1a gene promoter activation by GSK3b and associated factors”

Comment 6: Lithium treatment using young mice provided limited insights to the role of GSK3β in brain aging, which was still not addressed.

Response: The lithium treatment study was not designed to address the role of GSK3b in aging. Rather, we conducted those experiments to investigate whether the link between metabolism and GSK3b inhibition that we report in vitro might also be relevant in vivo. In addition to establishing that PGC-1a and its associated genes respond to GSK3b inhibition in vivo, we also showed that aging-associated patterns of gene expression are opposed by GSK3b inhibition. Although these data do not establish a role for GSK3b in brain aging, they are suggestive of a role therein, especially when taken together with the age-related increases in GSK3b protein abundance (Fig.S6).

Comment 7: Even from a brain metabolism perspective (not relevant to aging), data presented in this study provides limited new mechanistic insights comparing to the group's previous publication (Martin et al., 2018) except for the new RNAseq dataset and a list of enriched pathways.

Response: Our prior aging study (Martin et al., 2016) focused on imaging (histology, immunodetection, 2-photon redox imaging) and was limited to the hippocampal region only. Our next paper (Martin et al., 2018) focused on models of distinct brain cell types, H4 glioma and PC12 pheochromocytoma cells stimulated to differentiate into neuron-like cells. That study described metabolic differences among those cell types in response to GSK3b inhibition but, as we establish here, were a poor surrogate for primary cells. There, the focus was on the hippocampus, and analysis of the metabolic impact of lithium was limited to imaging-based measures. Our 2018 paper connected PGC-1a and GSK3b as intersecting aspects of growth regulation and metabolic status independent of age, and the 2016 paper indicated that aging was associated with metabolic adaptation.

The current study synthesizes and expands upon those earlier studies. We connect established inflammatory and neuronal dysfunction signatures of aging to changes in brain metabolism, a new insight. The demonstration that neurons have a specific promoter for PGC-1a expression that is unique to that cell type is another new insight: prior studies have only assigned it to "brain expression." The activation of this brain-specific promoter during neurogenesis has not previously been reported. The differential impact of GSK3b inhibition on each of the three promoters and the fact that there is cell-type specificity in the response of the promoters is also entirely new. The promoter-specific roles for CREB, AMPK, and TrkB in regulating PGC-1a isoform gene expression are also new. This paper delivers molecular-level insights on the regulation of brain energy metabolism that were not part of our prior publications and, to our knowledge, are not reflected in the public record to date.

Comment 8: The authors need to address the discrepancy in the effect of GSK3 β inhibition on neuronal / brain metabolism across two studies from the authors' group. In the present study, lithium was found to reduce neuronal mitochondrial respiration and potential, whereas in Martin et al., 2018, lithium was found to increase mitochondrial respiration and membrane potential.

Response: In the original submission, in the section "PGC-1a gene promoter activation by GSK3b and associated factors," we stated:

"In glioma cells (H4) and pheochromocytoma-differentiated neurons (PC12), GSK3b inhibition using lithium chloride (LiCl) causes an increase in expression of PGC-1a1 and PGC-1a4 isoform, but neither cell type expresses the B1E2 isoform which is the dominant form of PGC-1a in primary cortical neurons."

The key here is that the metabolic signature tracks with PGC-1a1 and PGC-1a4 in cells that do not express B1E2. The induction of PGC-1a1 and PGC-1a4 by lithium is a conserved feature across cell types, including primary neurons. As such, there is no discrepancy.

In the discussion, we also pointed out that "neuronal metabolic status aligns with expression from the brain-specific promoter, indicating a functional dominance of this isoform."

As our data show, lower expression of B1E2 coincided with lower values in metrics of mitochondrial activity, indicating that B1E2 is the functionally dominant isoform in neurons. In particular, suppression of B1E2 expression by lithium cannot be compensated for by lithium-induced increases in a1 and a4. We have added new sentences to the discussion.

Section: "Discussion," paragraph 3, lines 486 – 491.